# Graphitic phosphorus coordinated single Fe atoms for hydrogenative transformations

Xiangdong Long[1,2], Zelong Li[3], Guang Gao[1], Peng Sun[1], Jia Wang[1], Bingsen Zhang [4], Jun Zhong[5], Zheng Jiang [6] & Fuwei Li [1,7✉]

Single-atom metal-nitrogen-carbon (M-N-C) catalysts have sparked intensive interests, however, the development of an atomically dispersed metal-phosphorus-carbon (M-P-C) catalyst has not been achieved, although molecular metal-phosphine complexes have found tremendous applications in homogeneous catalysis. Herein, we successfully construct graphitic phosphorus species coordinated single-atom Fe on P-doped carbon, which display outstanding catalytic performance and reaction generality in the heterogeneous hydrogenation of N-heterocycles, functionalized nitroarenes, and reductive amination reactions, while the corresponding atomically dispersed Fe atoms embedded on N-doped carbon are almost inactive under the same reaction conditions. Furthermore, we find that the catalytic activity of graphitic phosphorus coordinated single-atom Fe sharply decreased when Fe atoms were transformed to Fe clusters/nanoparticles by post-impregnation Fe species. This work can be of fundamental interest for the design of single-atom catalysts by utilizing P atoms as coordination sites as well as of practical use for the application of M-P-C catalysts in heterogeneous catalysis.

[1] State Key Laboratory for Oxo Synthesis and Selective Oxidation, Suzhou Research Institute of LICP, Lanzhou Institute of Chemical Physics (LICP), Chinese Academy of Sciences, 730000 Lanzhou, China. [2] University of Chinese Academy of Sciences, 100049 Beijing, China. [3] Advanced Catalysis Center, College of Chemistry and Chemical Engineering, Lanzhou University, 730000 Lanzhou, China. [4] Shenyang National Laboratory for Materials Science (SYNL), Institute of Metal Research, Chinese Academy of Sciences, 110016 Shenyang, China. [5] Jiangsu Key Laboratory for Carbon-Based Functional Materials & Devices, Institute of Functional Nano & Soft Materials (FUNSOM), Soochow University, 215123 Suzhou, China. [6] Shanghai Synchrotron Radiation Facility, Zhangjiang Lab, Shanghai Advanced Research Institute, Chinese Academy of Science, 201210 Shanghai, China. [7] Dalian National Laboratory for Clean Energy, 116023 Dalian, China. ✉email: fuweili@licp.cas.cn

The coordination environment could deeply influence the catalytic performance of metal species through metal−support interactions (MSIs)[1,2], especially when the scale of metal species downsizes to single atom, that is, the single-atom catalysts (SACs)[3–5], wherein the metal atoms are located individually on support and are stabilized by neighboring surface atoms through covalent coordination or ionic interactions, and the coordination atoms could drastically tune the electronic and geometric structure of the supported single metal atoms[6,7]. Hence, modifying the surface atoms of support to construct an appropriate coordination environment for metal atoms is one of the most promising strategies to synthesize highly efficient and stable SACs[8,9]. Conventionally, the voids or vacancy defects on metal oxides (e.g., $TiO_2$, $CeO_2$, $ZrO_2$) or zeolites are commonly used as coordination sites for single metal atoms[10–12].

Over the past decades, the well-structured and crystallized nanocarbon-based materials have been widely used in catalysis, due to their unique electronic and structural properties as well as outstanding stability under hash reaction conditions[13]. Particularly, the carbon matrix could be easily modified via heteroatoms doping to introduce heteroatom-containing defects, which are ideal coordination sites for single metal atoms[14,15]. Within this context, metal−nitrogen−carbon catalyst (M−N−C, M refers to nonprecious Fe, Co, Ni etc.), using N to anchor metal atoms, have recently attracted tremendous attention as a representative bridging the homogeneous and heterogeneous catalysis[5], and the coordinated M−$N_x$ centers have displayed excellent activities in a vast array of thermal and electrocatalysis[16–19]. However, single-atom metal−phosphorus−carbon (M−P−C) catalyst with P atoms as the coordination sites has not been successfully established, which is principally due to the P atom having distinct physicochemical properties with N atom, such as larger atom radius and lower electronegativity, then doping the P atom into the carbon skeleton prefers to form a distorted configuration[20], and the P species are easily oxidized to electron-deficient $PO_x$ species, which will retard the formation of P-containing sites or defects to act as optimal electronic and geometric coordination environment for single metal atoms. In fact, P atom is widely used as the coordination center for metal in organometallic complexes[21], and it is well known that the molecular metal (particularly transition metal) phosphine complexes have found wide applications in homogeneous catalytic transformations, such as hydrogenations[22], coupling reactions[23,24], carbonylation[25,26], etc. Therefore, the development of single-atom M−P−C catalysts is in great demand for broadening SACs and exploring their sustainable catalysis.

Selective hydrogenative transformation of multifunctional substrates to valuable chemicals is a long-standing issue in catalysis. A prerequisite for achieving desired products with high selectivity is rationally designing the catalyst[27]. In this work, we report the fabrication of surface graphitic phosphorus species ($P_{grap}$)-coordinated Fe SAC (Fe-$P_{900}$-PCC) through templated-sacrificial approach using phytic acid and it contained Fe species as P and Fe precursors. It was revealed by integrated spectroscopy characterization and theoretical calculations that the Fe single atom is coordinated by four P atoms and one dioxygen molecule, adopting a pyramidal structure of $O_2$-Fe-$P_4$. After the dioxygen was removed by hydrogen under the reaction conditions, the in-situ-generated Fe-$P_4$ sites displayed high selectivity and outstanding stability for hydrogenations of different unsaturated functions (nitroarenes and N-heterocycles), and the reductive amination reactions as well. On the contrary, the planar Fe-$N_4$ active site in the corresponding Fe−N−C catalyst was inactive for these hydrogenative transformations under the same reaction conditions, suggesting that the coordinated P atoms play a crucial role for the catalytic performance of Fe single atom.

## Results and discussion

**Catalyst synthesis and characterization.** P-doped carbon catalysts are fabricated using a templated-sacrificial approach[28]. The typical synthesis process is illustrated in Fig. 1a. Briefly, the precursors (sucrose and phytic acid) and hard template (silica colloid, 14 nm $SiO_2$ nanoparticles dispersed in water) were mixed to form a homogeneous mixture and then carbonized at temperature between 700 and 1100 °C. Afterwards, the hard template was removed by hydrofluoric acid (10 wt%) to obtain porous carbon catalysts (PCCs). For comparison, non-heteroatom-doped PCC and N-doped PCC were also prepared in a similar procedure except that no phytic acid was added or phytic acid was replaced by cyanamide (N precursor), respectively. It is worth noting that the Fe element is ubiquitous in the earth's crust (the second most abundant metal); most commercial compounds or materials naturally contain low content of Fe impurities[29,30]. Directly utilizing raw materials containing metal as active components in the catalysts would be a quite meaningful strategy for the synthesis of catalysts[31,32]. Herein, we found that the Fe species originating from initial raw materials could be successfully inherited by the as-prepared carbon catalysts; no other external Fe salts are needed (the contents of Fe in raw materials are listed in Supplementary Table 1). The synthesized non-heteroatom-doped, N-doped and P-doped PCC are denoted as Fe-$C_x$-PCC, Fe-$N_x$-PCC and Fe-$P_x$-PCC, respectively ($x$ stands for the carbonization temperature in degree Celsius). The inductively coupled plasma mass spectroscopy (ICP-MS) analysis reveals Fe loading of 0.0016, 0.0023 and 0.071 wt% for Fe-$C_{900}$-PCC, Fe-$N_{900}$-PCC and Fe-$P_{900}$-PCC, respectively (Supplementary Table 2).

X-ray photoelectron spectroscopy (XPS) survey scans show that the heteroatoms (N or P) were doped into the porous carbon successfully (Supplementary Fig. 1). The elemental compositions of Fe-$C_{900}$-PCC, Fe-$N_{900}$-PCC and Fe-$P_{900}$-PCC are listed in Supplementary Table 3. The results of $N_2$ adsorption−desorption curves indicate that the synthesized catalysts possess an obvious mesoporous structure (Supplementary Fig. 2) and high specific Brunauer−Emmett−Teller (BET) surface area: Fe-$C_{900}$-PCC (768 $m^2 g^{-1}$), Fe-$N_{900}$-PCC (628 $m^2 g^{-1}$) and Fe-$P_{900}$-PCC (511 $m^2 g^{-1}$); structural properties of catalysts are listed in Supplementary Table 4. The high surface area and large pore volume are beneficial to the dispersion of active sites and the mass transfer of reactants. X-ray diffraction (XRD) patterns of catalysts are characterized by the low intensity of the broad peaks at $2\theta = 23$ and 43° (Supplementary Fig. 3), which are ascribed to the fingerprints of graphite (002) and (100) (ICSD No. 617290) reflections[33]. Moreover, no diffraction peaks corresponding to crystalline species of Fe are observed. Their Raman spectra (Supplementary Fig. 4) are dominated by two major peaks at 1338 $cm^{-1}$ (D-band) and 1589 $cm^{-1}$ (G-band) corresponding to the disordered and graphitic $sp^2$ carbon, respectively[34]. The transmission electron microscopy (TEM) images show that Fe-$C_{900}$-PCC, Fe-$N_{900}$-PCC and Fe-$P_{900}$-PCC have a sponge-like three-dimensional network structure (Fig. 1b and Supplementary Fig. 5). High-resolution TEM (HRTEM) images reveal that the bulk of catalysts is amorphous carbon, but the surface is surrounded by graphitic layers (4−7 layers, Fig. 1c and Supplementary Fig. 6). The existence of Fe in Fe-$N_{900}$-PCC and Fe-$P_{900}$-PCC was further confirmed by means of aberration-corrected scanning transmission electron microscopy (AC-STEM). As shown in Fig. 1d, Supplementary Figs. 7 and 8, single Fe atoms are dispersed on these catalysts. Additionally, some of the carbon-shells-encapsulated Fe nanoparticles are also observed in these carbon materials (Supplementary Figs. 9 and 10), but no Fe−Fe coordination has been detected in the characterization of Fe K-edge extended X-ray absorption fine structure (EXAFS, which will be discussed below), suggesting the majority of Fe

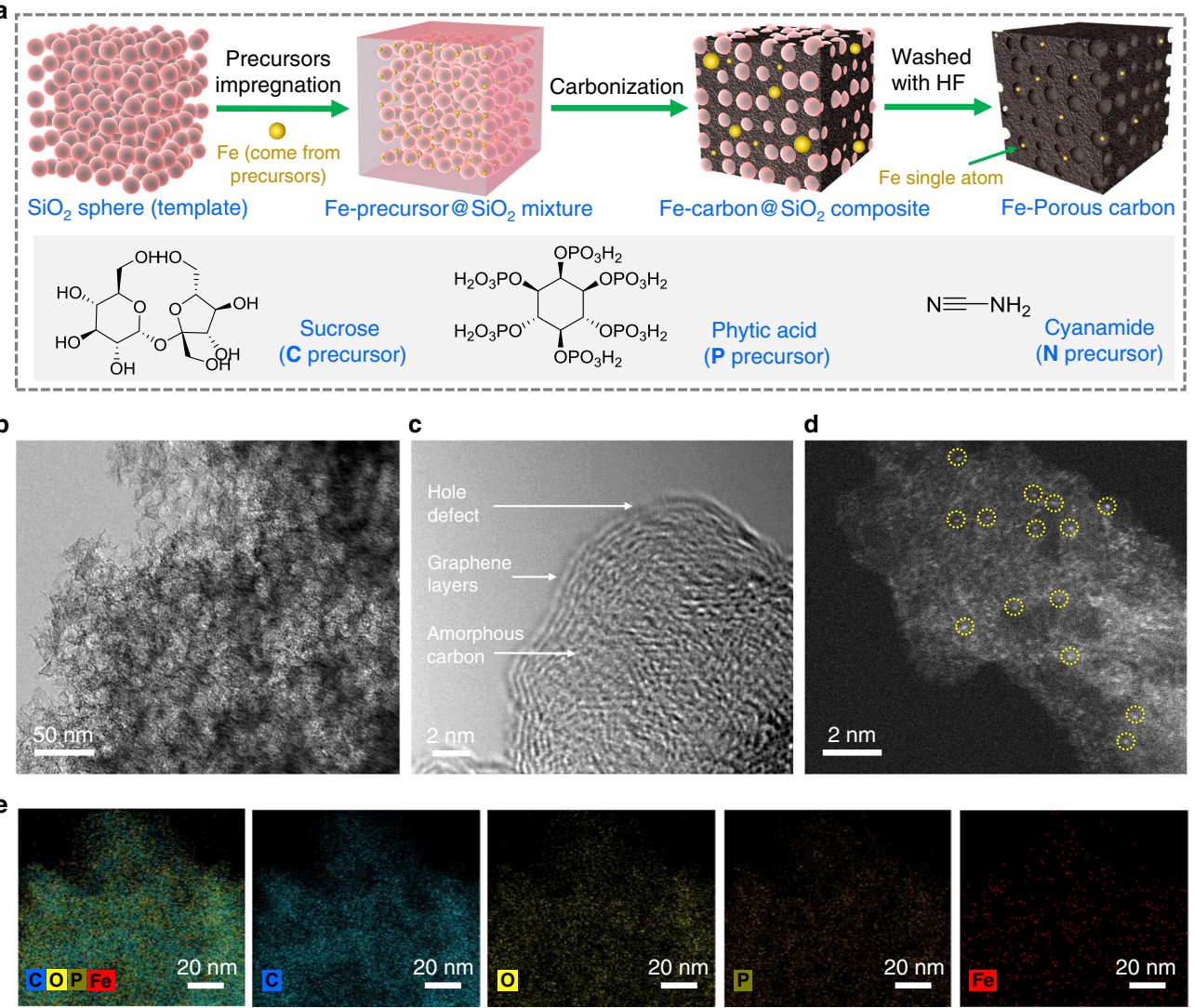

**Fig. 1 Structure characterizations of Fe-P$_{900}$-PCC. a** Schematic illustration of the preparation process of PCCs. **b** TEM image. **c** HRTEM image. **d** AC-STEM image, Fe single atoms are highlighted by yellow circles. **e** EDS mapping images for various elements.

species exist in the form of single atom, and the proportion of carbon-shells-encapsulated Fe aggregates is extremely low. For Fe-P$_{900}$-PCC, the energy-dispersive spectroscopy (EDS) mappings indicate that Fe, O and P are uniformly distributed on Fe-P$_{900}$-PCC (Fig. 1e).

**Catalytic performance**. Selective hydrogenation of unsaturated N-heterocycles plays an important role in the generation of aliphatic derivatives, which are crucial starting materials in the synthesis of pharmaceuticals, dyes and fine chemicals[35]. Here, the hydrogenation of quinoline is chosen as a model reaction to evaluate the catalytic performance of synthesized catalysts. As shown in Table 1, no quinoline is converted in the absence of catalyst (entry 1). The hydrogenation also does not occur over Fe-C$_{900}$-PCC catalyst (entry 2), even though the reaction temperature is raised to 200 °C (entry 3). Besides, Fe-N$_{900}$-PCC is also inactive at 150 °C (entry 4), but a 7% yield of tetrahydroquinoline has been achieved at 200 °C (entry 5). In the case of Fe-P$_{900}$-PCC catalyst, the conversion of quinoline reaches 93% at 150 °C with 99% of tetrahydroquinoline selectivity (entry 6). All the quinoline could be transformed to tetrahydroquinoline with prolonging reaction time to 18 h (entry 7), and no other by-products are detected by gas chromatograph (GC) and gas chromatograph-

mass spectrometer (GC-MS). The functional group tolerance survey of various unsaturated N-heterocycles was conducted with Fe-P$_{900}$-PCC as the optimal catalyst (Supplementary Fig. 11). The substituent position of quinoline has no influence on the yield of the corresponding tetrahydroquinoline (**2b−d**). Other functional groups, such as ester, hydroxyl, amino and benzo[h]quinoline, are also perfectly tolerated (**2e−h**). In addition, the oxygen-containing heterocycles, benzofuran, can also be hydrogenated to 2,3-dihydrobenzofuran (**2i**) under this catalytic system with 99% yield.

**Role of N and P**. The obvious differences of catalytic activity among Fe-C$_{900}$-PCC, Fe-N$_{900}$-PCC and Fe-P$_{900}$-PCC provoke us to discriminate the role of Fe, N and P in the PCC catalysts. Since the content of Fe in Fe-C$_{900}$-PCC and Fe-N$_{900}$-PCC is lower than that in Fe-P$_{900}$-PCC (Supplementary Table 2), we first wondered if the different content of Fe led to different catalytic activity, we then deliberately added 130 mg Fe(NO$_3$)$_3$·9H$_2$O into the initial precursor mixture of Fe-C$_{900}$-PCC and Fe-N$_{900}$-PCC to obtain high content of Fe samples, which are denoted as Fe@Fe-C$_{900}$-PCC and Fe@Fe-N$_{900}$-PCC. The ICP-MS results indicate that the content of Fe in Fe@Fe-C$_{900}$-PCC (0.038 wt%) and Fe@Fe-N$_{900}$-PCC (0.057 wt%) is much higher than the content of Fe in

**Table 1 Screening of catalysts for the hydrogenation of quinoline to tetrahydroquinoline.**

| Entry | Catalyst | Temperature (°C) | Conversion (%) | Yield (%) |
|---|---|---|---|---|
| 1 | Blank | 150 | 0 | 0 |
| 2 | Fe-C$_{900}$-PCC | 150 | 0 | 0 |
| 3 | Fe-C$_{900}$-PCC | 200 | 0 | 0 |
| 4 | Fe-N$_{900}$-PCC | 150 | 0 | 0 |
| 5 | Fe-N$_{900}$-PCC | 200 | 8 | 7 |
| 6 | Fe-P$_{900}$-PCC | 150 | 93 | 92 |
| 7[a] | Fe-P$_{900}$-PCC | 150 | 99 | 98 |
| 8 | Fe@Fe-C$_{900}$-PCC | 150 | 0 | 0 |
| 9 | Fe@Fe-N$_{900}$-PCC | 150 | 0 | 0 |
| 10 | P$_{900}$-PCC-polymer | 150 | 0 | 0 |
| 11 | Fe-P$_{900}$-PCC-polymer | 150 | 0 | 0 |

Reaction conditions: 1 mmol quinoline, 100 mg catalyst, 2 mL heptane, 4 MPa H$_2$, and 12 h. The conversion and yield are determined by GC using dodecane as internal standard.
[a]Reaction time is prolonged to 18 h.

Fe-C$_{900}$-PCC (0.0016 wt%) and Fe-N$_{900}$-PCC (0.0023 wt%), and is close to Fe-P$_{900}$-PCC (0.071 wt%). AC-STEM and HRTEM images as well as Fe K-edge EXAFS spectrum reveal that both Fe single atoms and nanoparticles are dispersed on Fe@Fe-C$_{900}$-PCC (Supplementary Fig. 12). However, this catalyst is inactive for hydrogenation of quinoline (Table 1, entry 8), suggesting that solely Fe-doped porous carbon without heteroatoms doping could not catalyze the hydrogenation reaction.

For Fe@Fe-N$_{900}$-PCC, the characterizations of AC-STEM and EXAFS demonstrate that the Fe atoms are atomically dispersed on N-doped porous carbon (Supplementary Fig. 13), and the quantitative EXAFS curve fitting reveals that the single Fe atom adopts a planar Fe-N$_4$ structure (see Supplementary Fig. 14 and Supplementary Table 6 for details). Surprisingly, Fe@Fe-N$_{900}$-PCC has no catalytic activity for hydrogenation (Table 1, entry 9), even though its Fe content is similar to the P-doped Fe-P$_{900}$-PCC. We then performed the density functional theory (DFT) calculations to investigate the catalytic behavior of Fe-N$_4$ sites toward H$_2$ adsorption and activation; the results demonstrate that H$_2$ can only physically adsorb on the Fe-N$_4$ site, and the resultant H–H bond length (0.75 Å) is equal to the free H$_2$ molecule (Supplementary Fig. 15), indicating that the Fe-N$_4$ sites are intrinsically inert to activate H$_2$, which is consistent with the experimental results. In fact, the Fe-N$_4$ site was mostly reported in oxidation reactions[16,19], electrochemical oxidation reduction reaction (ORR)[36,37] or CO$_2$ electroreduction reaction[38,39], and Fe-N-C SACs for hydrogenation reaction is rarely reported.

Next, we investigated the reason why Fe-P$_{900}$-PCC has high activity for hydrogenation reaction. Firstly, in order to explore whether the Fe-free P-doping carbon catalyst has the activity for hydrogenation[40,41], we particularly synthesized an ultrapure P-doped carbon (denoted as P$_{900}$-PCC-polymer) via pyrolysis of polytris(4-vinylphenyl)phosphane, which is synthesized according to Ding's reported procedure[42], and its Fe-containing analog (denoted as Fe-P$_{900}$-PCC-polymer) was also prepared as a reference catalyst through pyrolysis of Fe-coordinated polytris (4-vinylphenyl)phosphane. ICP-MS analysis reveals Fe loading of 0.000023 and 0.096 wt% for P$_{900}$-PCC-polymer and Fe-P$_{900}$-PCC-polymer, respectively. Nevertheless, both P$_{900}$-PCC-polymer and Fe-P$_{900}$-PCC-polymer showed no activity in the quinoline hydrogenation (Table 1, entries 10 and 11). One possible reason is that the surface area of these polymer-derived catalysts is extremely low (<1 m$^2$ g$^{-1}$); they are not suitable for the liquid-

phase catalytic reaction. Therefore, we then performed the gas-phase isotopic H$_2$-D$_2$ exchange experiment to investigate the dissociation of H$_2$ on P$_{900}$-PCC-polymer and Fe-P$_{900}$-PCC-polymer. As shown in Supplementary Fig. 16a, under the continuous flow of H$_2$ and D$_2$, no HD (m/z = 3) signal is detected over P$_{900}$-PCC-polymer when the temperature is below 300 °C. On the contrary, HD is formed starting at 100 °C over Fe-P$_{900}$-PCC-polymer, similar to the HD profile of Fe-P$_{900}$-PCC (Supplementary Fig. 16b), suggesting that the present P-doped and Fe-free carbon catalyst cannot activate H$_2$, and the addition of Fe boosts the activation and dissociation of H$_2$. Since above investigations have revealed that the solely Fe species was inactive for hydrogenation reaction (Fe-C$_{900}$-PCC and Fe@Fe-C$_{900}$-PCC, Table 1, entries 2, 3, and 8), so we speculated that Fe species and P species in Fe-P$_{900}$-PCC co-catalyzed the H$_2$ dissociation and hydrogenation reaction.

**Origin of catalytic activity**. To probe the underlying interplay between P and Fe species, a series of Fe-P$_x$-PCCs ($x$ stands for carbonization temperature, from 700 to 1100 °C) were synthesized by varying the carbonization temperature. The P contents of Fe-P$_{700}$-PCC, Fe-P$_{800}$-PCC, Fe-P$_{900}$-PCC, Fe-P$_{1000}$-PCC and Fe-P$_{1100}$-PCC are 2.04, 3.48, 2.74, 2.04 and 0.91 atomic %, respectively, determined by XPS. The state of P was initially investigated by analyzing P 2p XPS core level peaks of Fe-P$_x$-PCCs. As shown in Fig. 2a, P 2p XPS spectrum of Fe-P$_{700}$-PCC could be deconvoluted into two bands: C-O-P (134.4 eV) and C-PO$_3$/C$_2$-PO$_2$ (133.1 eV), which are denoted as PO$_x$, representing these P atoms are in a high oxidation state[43,44]. Remarkably, a new band in the less oxidized state appears at 132.1 eV for Fe-P$_{800}$-PCC, Fe-P$_{900}$-PCC, Fe-P$_{1000}$-PCC and Fe-P$_{1100}$-PCC, which are assigned to graphitic P group (P$_{grap}$) generated by substitution of a graphitic C atom in the framework of graphite with a P atom[45]. The percentages of P$_{grap}$ in Fe-P$_{700}$-PCC, Fe-P$_{800}$-PCC, Fe-P$_{900}$-PCC, Fe-P$_{1000}$-PCC and Fe-P$_{1100}$-PCC are 0%, 6.0%, 15.0%, 17.4% and 22.7%, respectively, indicating that P$_{grap}$ cannot be formed at lower carbonization temperature (700 °C), while the higher carbonization temperature will promote more P atoms to insert into the skeleton of graphite forming high percentage of P$_{grap}$ structure.

These results were further confirmed by solid-state $^{31}$P NMR spectroscopy (Fig. 2b). $^{31}$P NMR spectrum of Fe-P$_{700}$-PCC exhibits a broad signal at −1.95 ppm, which is attributed to

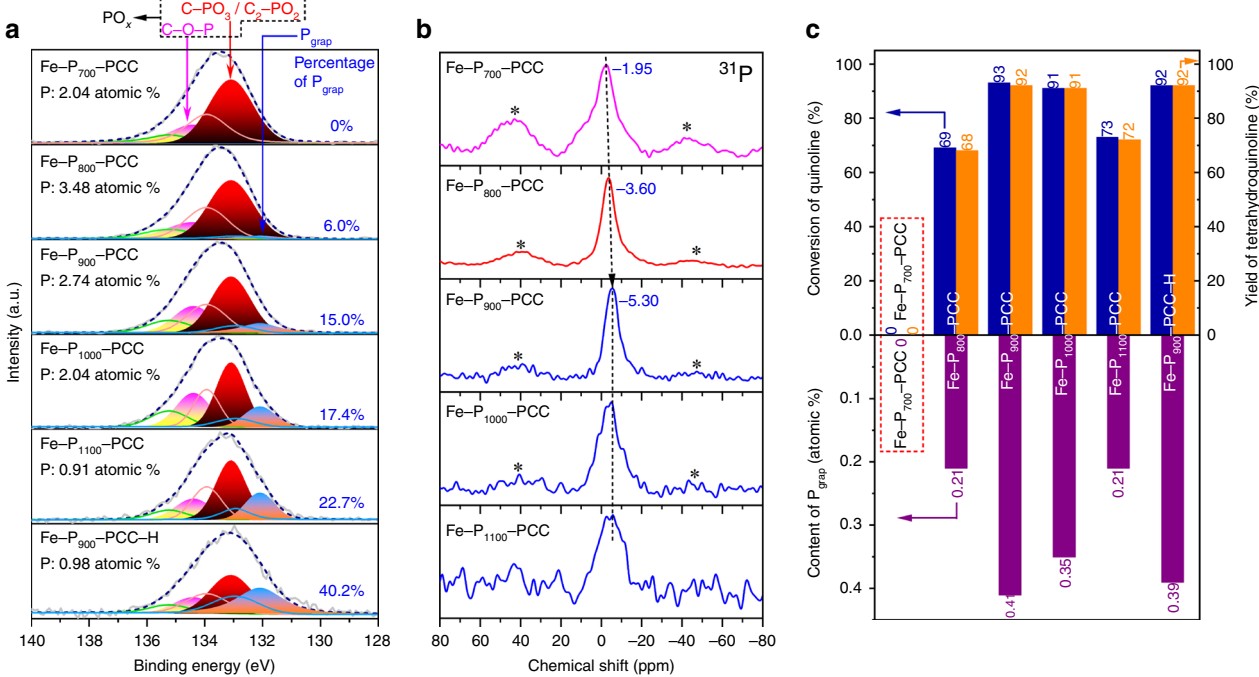

**Fig. 2 Characterizations of P species. a** Deconvoluted P 2p XPS spectra of Fe-$P_x$-PCCs. All of the spectra were deconvoluted using three doublet peaks with an area ratio of 0.5 and a separation between peaks of 0.84 eV. **b** Solid-state $^{31}P$ NMR spectra of Fe-$P_x$-PCCs. Spinning side bands are marked with asterisk. **c** The relationships between catalytic activity and content of $P_{grap}$.

organic phosphonate with high oxidation state[46]. The main peak for Fe-$P_{800}$-PCC shifts to −3.60 ppm, indicating a small part of $P_{grap}$ species tend to form, since the signal of $P_{grap}$ is around −5 ppm[44,47]. For the spectra of Fe-$P_{900}$-PCC, Fe-$P_{1000}$-PCC and Fe-$P_{1100}$-PCC, the main peak is central at −5.30 ppm, which shows a relatively high percentage of $P_{grap}$ has formed in these catalysts. These results are in good consistence with the P 2p XPS results.

To determine which type of P species is responsible for promoting the catalytic activity, hydrogenations of quinoline catalyzed by Fe-$P_x$-PCCs were conducted (Fig. 2c) and it is found that their catalytic activities are positively associated with the content of $P_{grap}$ (Fig. 2c, content of $P_{grap}$ (atomic %) = total content of P (atomic %) × percentage of $P_{grap}$ (%)): there is no $P_{grap}$ in Fe-$P_{700}$-PCC, and it is inactive for hydrogenation; for other four samples, the content of $P_{grap}$ follows the order of Fe-$P_{900}$-PCC > Fe-$P_{1000}$-PCC > Fe-$P_{1100}$-PCC > Fe-$P_{800}$-PCC; this trend is exactly consistent with the variation tendency of the conversion of quinoline, Fe-$P_{900}$-PCC (93%) > Fe-$P_{1000}$-PCC (91%) > Fe-$P_{1100}$-PCC (73%) > Fe-$P_{800}$-PCC (69%). In contrast, there is no correlation between the catalytic activity and the content of $PO_x$ (listed in Supplementary Table 5). Furthermore, when Fe-$P_{900}$-PCC was treated with $H_2$ at 800 °C for 2 h (named as Fe-$P_{900}$-PCC-H), the total P content decreases from 2.74 atomic % in Fe-$P_{900}$-PCC to 0.98 atomic % in Fe-$P_{900}$-PCC-H, mainly due to the decrease of unstable $PO_x$ species (C-O-P and C-$PO_3$/$C_2$-$PO_2$), but the content of $P_{grap}$ is consistent (0.41 vs 0.39 atomic %, Fig. 2a). Interestingly, Fe-$P_{900}$-PCC-H shows equal catalytic activity in the hydrogenation of quinoline with that of the Fe-$P_{900}$-PCC (Fig. 2c). All of these findings demonstrate that $P_{grap}$ plays an essential role in the hydrogenation reaction.

Next, the information of Fe species in Fe-$P_x$-PCCs was investigated. The results of ICP-MS reveal that only a relatively few amounts of Fe remain in synthesized porous carbon (0.0072 wt%) at lower carbonization temperature (Fe-$P_{700}$-PCC); higher carbonization temperature facilitates more Fe doped into the resultant PCCs: Fe-$P_{800}$-PCC (0.035 wt%), Fe-$P_{900}$-PCC (0.071 wt%), Fe-$P_{1000}$-PCC (0.11 wt%), Fe-$P_{1000}$-PCC (0.068 wt%). EXAFS was used to analyze the coordination environments around Fe atoms in Fe-$P_x$-PCCs. Figure 3a shows the Fourier transform curves at Fe K-edge of Fe-$P_x$-PCCs. For Fe-$P_{800}$-PCC, the first shell peak is at 1.53 Å, which slightly deviates from the coordination of Fe-O (1.45 Å) and might be a small part of Fe-P coordination is formed. Besides, Fe−Fe coordination could be clearly observed, indicating the formation of Fe aggregates. After raising the carbonization temperature to 900 °C, a shell peak at 1.63 Å is detected, which is obviously distinct from Fe−Fe (2.21 Å) and Fe−O coordination and could be attributed to the Fe−P coordination based on refs. [48,49]. Furthermore, there is no Fe−Fe coordination peak in the Fe-$P_{900}$-PCC, suggesting all Fe species on the surface are atomically dispersed. A predominant peak associated with Fe−P and a weak intense feature attributed to Fe −Fe (3.5 Å) are simultaneously observed in Fe-$P_{1000}$-PCC and Fe-$P_{1100}$-PCC, demonstrating that their surface Fe species are mainly atomically dispersed Fe alongside some Fe aggregates.

$^{57}Fe$ Mössbauer spectroscopy is a powerful technique to investigate the coordination and valence state of Fe species in the solid state[19,50]. The best catalyst, Fe-$P_{900}$-PCC, was studied by Mössbauer spectroscopy at room temperature. As shown in Fig. 3b, the spectrum of Fe-$P_{900}$-PCC is well described by a single doublet with isomer shift (IS) value of 0.31 mm s$^{-1}$ and quadrupole splitting (QS) value of 0.66 mm s$^{-1}$, no sextet and singlet are detected, indicating only one type of Fe species exists in Fe-$P_{900}$-PCC and the absence of crystalline Fe species. This result is well consistent with the EXAFS result revealing no Fe nanoparticles are present on the surface of Fe-$P_{900}$-PCC, except for single Fe atoms. The value of IS is small and lies in the typical range of $Fe^{3+}$ low-spin complexes, suggesting Fe exists in the form of $Fe^{3+}$ in Fe-$P_{900}$-PCC[51], which is also consistent with its Fe 2p XPS result (Supplementary Fig. 17). Furthermore, the values of IS and QS agree with Mössbauer parameters of organic P-ligand coordinated Fe complexes[52,53], indicating atomically

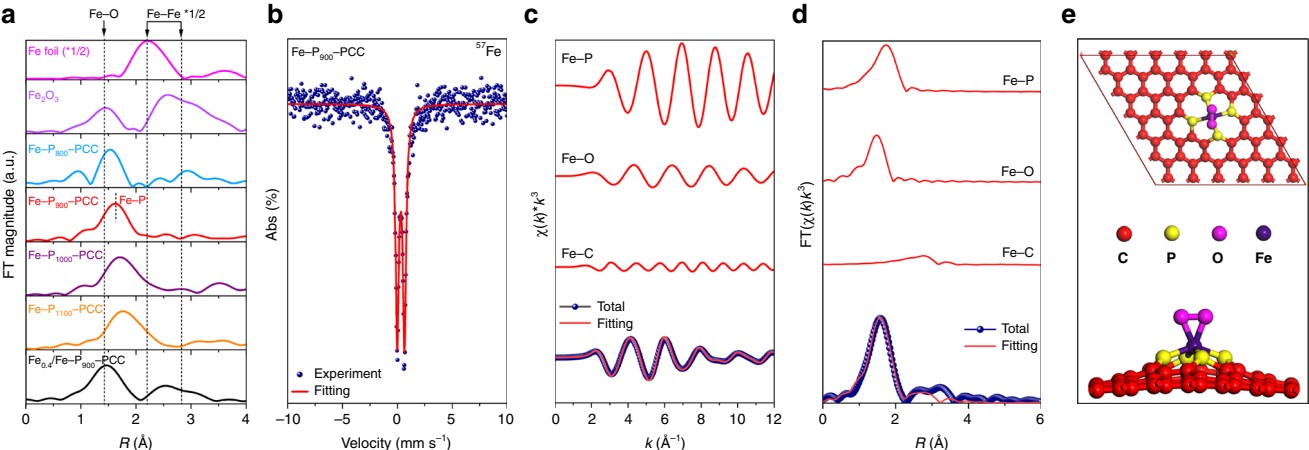

**Fig. 3 Characterizations of Fe species. a** EXAFS spectra of Fe-P$_x$-PCC, Fe$_{0.4}$/Fe-P$_{900}$-PCC and reference materials (Fe foil and Fe$_2$O$_3$). **b** $^{57}$Fe Mössbauer spectrum of Fe-P$_{900}$-PCC. **c**, **d** The experimental Fe K-edge EXAFS data (blue dot) and the fitting curve (red line) of Fe-P$_{900}$-PCC at $k$ and R space, respectively. **e** Top and side views of optimized structural model of O$_2$-Fe-P$_4$.

dispersed Fe atoms on the surface of Fe-P$_{900}$-PCC are coordinated by surface P$_{grap}$ groups.

The coordination information of Fe atom in optimal Fe-P$_{900}$-PCC is further studied using quantitative EXAFS curve fitting (see Supplementary Table 6 for details). The best-fitting analyses are shown in Fig. 3c, d, which manifest that the Fe atoms are coordinated by four P atoms and a dioxygen molecule (O$_2$-Fe-P$_4$), forming a pyramidal geometry as shown in Fig. 3e; this structure is quite different from the planar structure of Fe-N$_4$. Besides, the theoretical calculations reveal that the binding energy of the O$_2$-Fe-P$_4$ structure is −2.09 eV, suggesting this coordination structure is thermodynamically stable.

The formation and evolution of P species and Fe species in Fe-P$_x$-PCC, and their relationships with catalytic activity are proposed as below. At 700 °C, P atoms cannot insert into carbon framework to form P$_{grap}$; all of the P species are electron-deficient PO$_x$ with high oxidation state, which cannot form stable Fe-P structure with Fe species; the vast majority of Fe that came from raw materials were removed by hydrofluoric acid at template-removing step, so Fe-P$_{700}$-PCC is inactive at all for hydrogenation; a relatively low content of P$_{grap}$ is formed in Fe-P$_{800}$-PCC (0.21 atomic %), and some of Fe atoms are coordinated by P$_{grap}$ to form O$_2$-Fe-P$_4$, the conversion of quinoline raised to 69%; when carbonization temperature was enhanced to 900 °C, the content of P$_{grap}$ reached the maximum, and the formation of Fe aggregates is significantly prevented because of the coordination between Fe and the P$_{grap}$ species, which then facilitate the construction of atomically dispersed O$_2$-Fe-P$_4$ sites on the P-doped porous carbon, and the best catalytic activity was obtained with this optimal Fe-P$_{900}$-PCC catalyst; the content of P$_{grap}$ decreased (0.41−0.35, 0.21 atomic %, respectively) when further increasing the carbonization temperature to 1000 and 1100 °C, and a part of O$_2$-Fe-P$_4$ is aggregated to Fe clusters or nanoparticles, so the conversions of quinoline decreased to 91% and 73%, respectively. All these investigations indicate that the P species have a significant impact on the evolution of Fe species; the formation of P$_{grap}$ is the prerequisite for the existence of O$_2$-Fe-P$_4$ on the surface of P-doped porous carbon, and O$_2$-Fe-P$_4$ sites are responsible for H$_2$ dissociation and hydrogenation reaction.

In order to further prove that P$_{grap}$-coordinated single-atom Fe is the active site for hydrogenation reaction, we used the reverse verification method. Chemical titration to selectively poison active sites would be quite useful to achieve this goal[54]; herein, we intended to convert the P$_{grap}$-coordinated single Fe atoms to Fe

clusters or nanoparticles via impregnating Fe species on Fe-P$_{900}$-PCC. Subsequently, four Fe$_x$/Fe-P$_{900}$-PCC ($x$ stands for the content of impregnated Fe, wt%) samples with different Fe content (0.11, 0.20, 0.40, 0.95 wt%, determined by ICP-MS) were synthesized by wet impregnation method using Fe-P$_{900}$-PCC as the support. Surprisingly, the conversions of quinoline over these Fe$_x$/Fe-P$_{900}$-PCC catalysts drastically decreased to 19%, 13%, 7%, 5%, respectively (Supplementary Table 7).

Then, the difference of surface chemical state between Fe-P$_{900}$-PCC and Fe$_x$/Fe-P$_{900}$-PCC has been investigated. The EXAFS spectra of Fe-P$_{900}$-PCC and Fe$_{0.4}$/Fe-P$_{900}$-PCC show that the original Fe-P coordination in Fe-P$_{900}$-PCC is replaced by Fe−Fe and Fe−O coordination after external Fe sources were impregnated on the surface of Fe-P$_{900}$-PCC (Fig. 3a), which suggests that P$_{grap}$-coordinated single Fe atoms have been transformed to Fe aggregates. On the other hand, the main peak of P 2p XPS spectra of Fe$_x$/Fe-P$_{900}$-PCC has shifted to higher binding energy after introducing Fe species into Fe-P$_{900}$-PCC, and the shifts become more obvious with the increase of Fe loading (Supplementary Fig. 18), which suggests that the electronic state of P species has been changed by the post-impregnation Fe, and these changes of Fe species and P species lead to sharp decreases in catalytic activity, reversely proving that P$_{grap}$-coordinated single-atom Fe is the active site of this Fe-P-C catalyst.

**DFT calculations**. To understand the possible mechanism for the hydrogenation of quinoline, we studied the reaction process with DFT calculations, which initially reveals that the bond length of O−O in O$_2$-Fe-P$_4$ is 1.314 Å; the two oxygen atoms have a total magnetic moment of 0.77 μB, O$_2$ gains approximately one electron from Fe-P$_4$ upon adsorption, and the adsorbed O$_2$ is assigned as O$_2^−$ which can be readily reduced by H$_2$ since the reaction (O$_2$-Fe-P$_4$ + 2H$_2$ → Fe-P$_4$ + 2H$_2$O) is exothermic by 4.08 eV. Therefore, the Fe-P$_4$ was used as the starting point to calculate the reaction pathway of hydrogenation of quinoline. As shown in Fig. 4, the H$_2$ preferentially adsorbs on Fe atom, the bond length of H−H is elongated to 0.81 Å with a free energy of −0.407 eV, indicating that the H$_2$ can easily adsorb and bond on Fe-P$_4$. On the other hand, the adsorption energy of quinoline on Fe atoms is −0.687 eV (Supplementary Tables 8 and 9). Then, one hydrogen is transferred to N atom (C$_9$H$_8$N*), leaving another hydrogen bound to Fe atom (int-2 to int-3). Hydrogenation of C$_9$H$_8$N* to adsorbed C$_9$H$_9$N* proceeds with an energy barrier of 0.38 eV and is exothermic by −0.348 eV (from int-3 to int-4). The

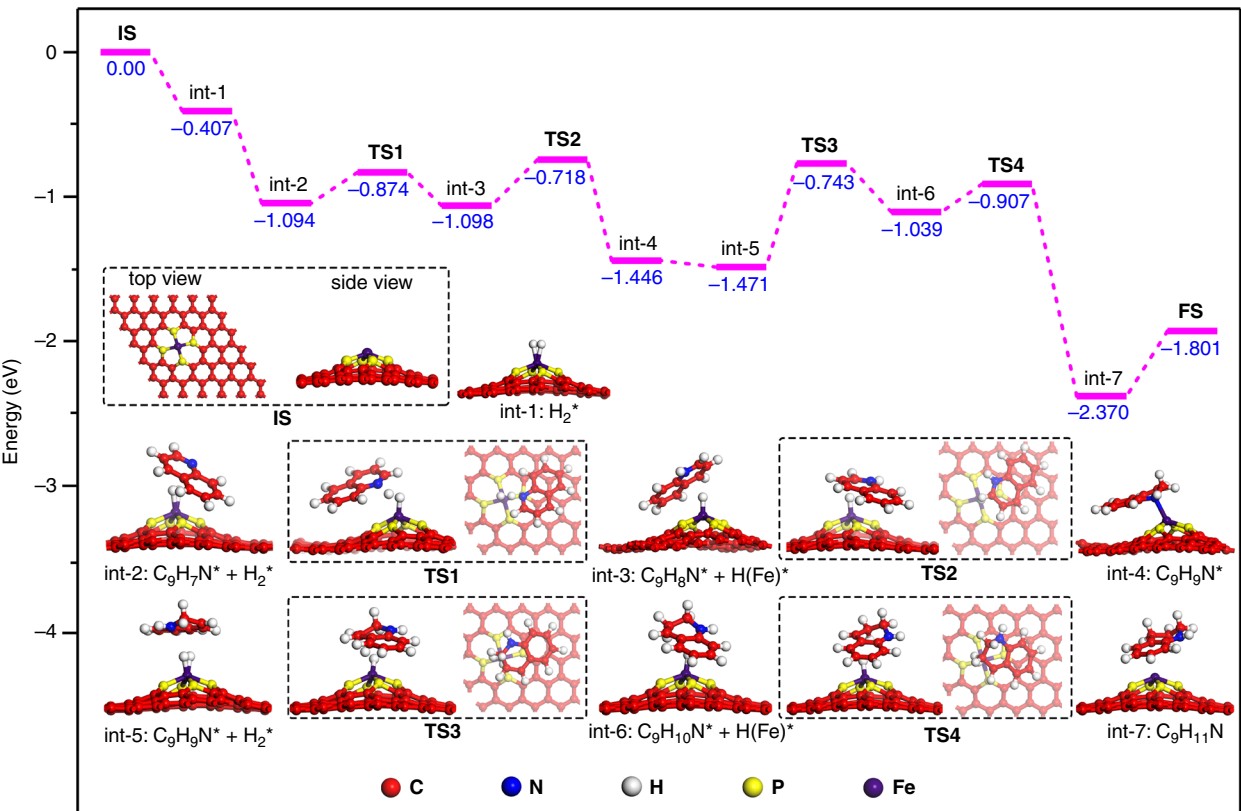

**Fig. 4 Catalytic mechanism study of Fe-P$_{900}$-PCC for hydrogenation of quinoline.** Energies of intermediates (int) and transition states (TS) in the mechanism of quinoline stepwise hydrogenation on the O$_2$-Fe-P$_4$ from DFT calculations. IS initial state, FS final state.

reaction continues to give the final product via the addition of second hydrogen molecule, and the reaction of C$_9$H$_9$N* to C$_9$H$_{10}$N* (int-5 to int-6) is found to be the rate-determining step with an energy barrier of 0.728 eV.

**Application of Fe-P$_{900}$-PCC.** Amines are a privileged class of compounds used extensively in bulk and fine chemicals, pharmaceuticals, and materials[55]. Catalytic reductive amination of aldehydes and/or ketones with molecular hydrogen represents one of the most attractive methods for the preparation of advanced amines. Conventional synthesis relied on noble metal-based homogeneous or heterogeneous catalysts[56,57]. Very recently, Co or Ni nanoparticle catalysts encapsulated by N-doped carbon shells derived from pyrolysis of metal organic complex have shown outstanding applications for this reaction[58,59]. To the best of our knowledge, the SACs have not been successfully applied in this transformation. Herein, the optimized Fe-P$_{900}$-PCC catalyst demonstrates excellent selectivity and activity in the synthesis of primary, secondary and tertiary amines, including some chiral amines and drug targets. As shown in Fig. 5a, a variety of alkylamines (methylamine, *n*-propylamine, *n*-hexylamine and piperidine) could react with benzaldehyde to form corresponding secondary amines (**5a−d**) with excellent yields (>89%). Moreover, ethanolamine and 2,6-dimethylaniline are also compatible with this catalytic system (**5e**, **5f**). With methylamine as a standard amine, the as-prepared Fe-P$_{900}$-PCC catalyst also showed good scope tolerance to different aldehydes and ketones, affording the desired secondary amines in 90−98% yields (**5g−l**). Gratifyingly, the original stereo configuration of the chiral amine could be perfectly retained over the Fe-P$_{900}$-PCC catalyst. The reductive alkylation of representative (R)-1-pheny-lethan-1-amine and (S)-1-phenylethan-1-amine gave the

expected chiral secondary amines in 83% yield with 97% ee (**5m−o**). Moreover, Fe-P$_{900}$-PCC catalyst also demonstrated good compatibility for the synthesis of tertiary amine, taking Buclizine (**5p**) as a representative target, which is a drug for the prevention and treatment of nausea, vomiting and dizziness associated with motion sickness[60], could be facilely prepared in eight gram scale with 76% isolated yield via this catalytic system.

4-(aminomethyl)benzoic acid (**5q**) is an important hemostatic drug and a functional primary amine as well[61]. Currently, its industrial synthesis relies on Raney Ni, which is still facing challenges in low selectivity and metal residues[62], and a large amount of metal chelating agents (usually ethylenediaminete-traacetic acid, EDTA) and water are needed to remove Ni residue to parts-per-million (ppm) level. To our delight, Fe-P$_{900}$-PCC catalyst showed excellent activity and selectivity in the reductive amination of 4-formylbenzoic acid (7.05 g) with aqueous ammonia in water under mild reaction temperature (75 °C), almost quantitative yield (98%) of the desired **5q** was obtained without any other products observed. ICP-MS test revealed that there is almost no metal contaminations remained in the final aqueous reaction solution (Fe, Co, Ni, Cu < 10 ppb), while the Ni residue was detected to be as high as 28,653 ppb when Raney Ni is used as the catalyst under the same reaction conditions, demonstrating the great potential of present Fe-P$_{900}$-PCC in the synthesis of primary amines, particularly for medical reagent.

Functionalized anilines are important intermediates or building blocks in chemical industry[63], and heterogeneous hydrogenation of nitroarenes with H$_2$ to prepare anilines has been extensively studied by supported nanoparticle catalysts[64,65] and few SACs (e.g., Pt$_1$/FeO$_x$[66], Pt$_1$/α-MoC[67]), but to date the Fe-based SAC has not been investigated. Similarly, the main challenge for this transformation is the selective hydrogenation of functional nitroarenes bearing another moiety that can

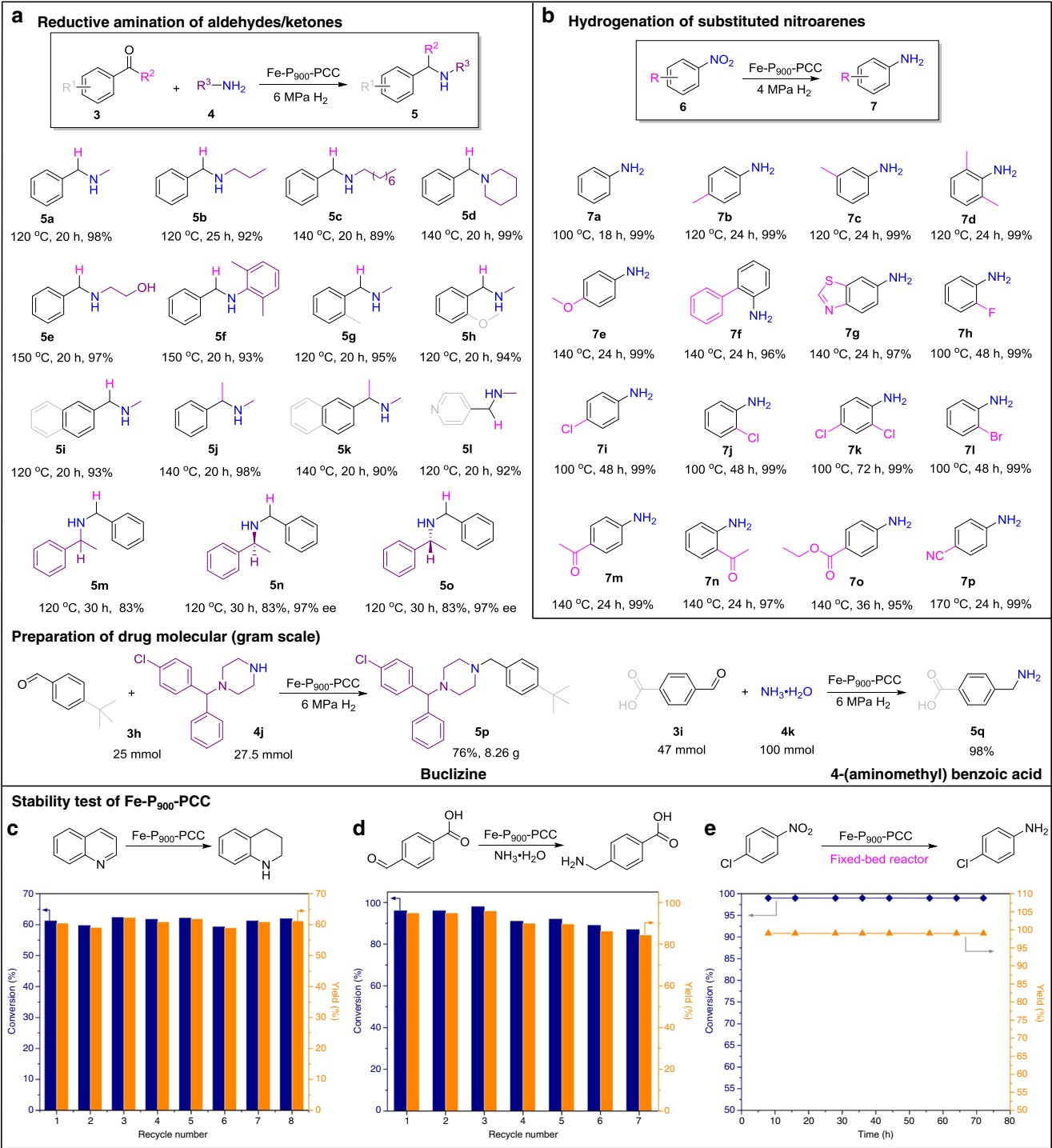

**Fig. 5 Fe-P$_{900}$-PCC catalyzed reductive amination and hydrogenation reactions. a** Reaction conditions: 2 mmol aldehydes/ketone, 40 mg Fe-P$_{900}$-PCC, 3 mmol amine, 5 mL ethanol, 6 MPa H$_2$. Yields are determined by GC using 1,4-dioxane as an internal standard. Reaction conditions of gram scale experiments: for synthesis of **5p**, 25 mmol **3h**, 27.5 mmol **4j**, 500 mg Fe-P$_{900}$-PCC, 30 mL ethanol, 6 MPa H$_2$, 120 °C, 30 h, isolated yield; for synthesis of **5q**, 47 mmol **3i**, 15 mL aqueous ammonia (100 mmol NH$_3$), 15 mL H$_2$O, 700 mg Fe-P$_{900}$-PCC, 6 MPa H$_2$, 75 °C, 30 h, yield is determined by high performance liquid chromatography (HPLC). **b** Reaction conditions: 1 mmol substrate, 100 mg Fe-P$_{900}$-PCC, 4 MPa H$_2$, 2 mL toluene, yields are determined by GC (*n*-hexadecane as an internal standard). Stability test of Fe-P$_{900}$-PCC. **c** Reaction conditions: 1 mmol quinoline, 50 mg Fe-P$_{900}$-PCC, 2 mL heptane, 4 MPa H$_2$, 150 °C, 12 h. Recovered catalyst is washed by ethanol, dried at 60 °C, and then submitted to the next batch of reaction. **d** Reaction conditions: 2 mmol substrate, 2 mL aqueous ammonia, 40 mg Fe-P$_{900}$-PCC, 3 mL H$_2$O, 6 MPa H$_2$, 75 °C, 20 h. Recovered catalyst is washed by ethanol, dried at 60 °C, and then submitted to the next batch of reaction. **e** Long-term test of Fe-P$_{900}$-PCC for hydrogenation of 4-chloronitrobenzene. Reaction conditions: 4 MPa H$_2$, 100 °C, H$_2$ (50 mL min$^{-1}$) and 10 wt% 4-chloronitrobenzene in toluene (0.05 mL min$^{-1}$) were introduced into the reactor.

possibly be hydrogenated too[68,69]. As shown in Fig. 5b, the optimized Fe-P$_{900}$-PCC is not only highly active for the hydrogenation of normal nitroarenes and nitroheteroarene to yield the corresponding anilines with almost quantitative yields (**7a−g**), but also highly selective for hydrogenative transformations of functional nitroarenes that possess different halogens, ketone, ester and nitrile substituents, where the side dehalogenation or undesirable hydrogenations easily occur; only nitrogroups have been selectively reduced and excellent yields (95−99%) of expected functional anilines (**7h−p**) were obtained.

**Stability test**. The excellent catalytic performance of Fe-P$_{900}$-PCC is comparable or outperformances a majority of previously reported hydrogenation catalysts (Supplementary Tables 10−12). Then, the stability of Fe-P$_{900}$-PCC was first studied in the hydrogenation of quinoline in stirring autoclave; it demonstrated high stability under a controlled conversion of about 60% and could be conveniently recycled up to eight times without any deactivation (Fig. 5c). Furthermore, even under the harsh experimental conditions (aqueous ammonia solution, in the reaction of reductive amination of 4-formylbenzoic acid), the present Fe-P$_{900}$-PCC displayed a just slight loss of catalytic activity within seven cycles (Fig. 5d). To investigate the generality potential of Fe-P$_{900}$-PCC both in different hydrogenations and reactors conditions as well, we then tested the catalyst lifetime in a fixed-bed reactor with the hydrogenation of 4-chloronitrobenzene as a model reaction, as shown in Fig. 5e. Fe-P$_{900}$-PCC could stably run for 72 h without activity loss. As revealed in the literatures that the coordination environment is crucial to the stability of the highly dispersed catalyst and SAC[8,70], here the ICP-MS, XPS, EXAFS and STEM were employed to confirm the chemical stability of Fe-P$_{900}$-PCC. ICP-MS analysis reveals that the content of Fe in the spent Fe-P$_{900}$-PCC is almost the same with the fresh catalyst (0.069 vs 0.071 wt%). P 2p and Fe 2p XPS spectra reveal that the chemical state of P$_{grap}$ and Fe species is constant (Supplementary Fig. 19a, b), the content of P$_{grap}$ remains unchanged at 0.41 atomic% before and after hydrogenations (Supplementary Table 5). The AC-STEM image and Fe K-edge EXAFS spectrum of spent Fe-P$_{900}$-PCC indicate that the atomically dispersed Fe species are well preserved after eight repetitive runs (Supplementary Fig. 19c, d).

In summary, we have facilely developed an atomically dispersed Fe-P-C catalyst (Fe-P$_{900}$-PCC). Systematic characterizations and control experiments prove that the single Fe atom and the surface P$_{grap}$ species formed a unique distorted O$_2$-Fe-P$_4$ structure on the P-doped porous carbon. After the dioxygen of O$_2$-Fe-P$_4$ was reduced by H$_2$ under reaction conditions, the in-situ-generated Fe-P$_4$ sites displayed outstanding catalytic activity, selectivity and substrate generality in hydrogenations and reductive amination reactions to produce a wide variety of primary, secondary and tertiary amines, including drug targets. On the contrary, the planar Fe-N$_4$ active site in the corresponding Fe-N-C catalyst was inactive for hydrogenations under the same reaction conditions. Moreover, the Fe-P$_{900}$-PCC exhibits remarkable hydrogenation stability in the autoclave and fixed-bed reactors. Our work provides the indication of heterogeneous M-P-C SACs, which will broaden the fabrication of SACs and inspire their applications in heterogeneous catalysis.

## Methods

**Materials**. Sucrose (ultrapure, 99.9%) was purchased from Shanghai Macklin Biochemical Co. Ltd. Cyanamide (98%) and silica colloid (nanoparticle dispersed in water, 40 wt%, particle size (14 nm), surface area (250 m$^2$ g$^{-1}$)) were purchased from Alfa Aesar (China) Chemicals Co. Ltd and Adamas Reagent Ltd. Phytic acid solution (50 wt% in H$_2$O) was purchased from Aladdin Industrial Cooperation. Hydrofluoric acid (40 wt%) was purchased from Shanghai Chemical Reagent Co. Ltd. All of the chemical reagents were used as received without further purification.

Ultrapure water (18.25 MΩ cm) was obtained from the Molecular Molelement 1810A system.

**Synthesis of Fe-P$_{900}$-PCC**. Sucrose (12.5 g) and phytic acid solution (5 g) were dissolved into 50 g ultrapure water, followed by the addition of 20 g silica colloid; the mixture was stirred at room temperature for 12 h to obtain a homogeneous slurry. Then the mixture was placed in a drying oven at 100 °C for 8 h and 160 °C for another 8 h. The obtained black lumps were grounded into fine powder and then transferred into a tube furnace for carbonization under the Ar atmosphere. The temperature was controllably ramped at a rate of 2 °C min$^{-1}$ to 600 °C, and subsequently 5 °C min$^{-1}$ to 900 °C, finally maintained at 900 °C for 3 h. When cooling to room temperature, the obtained carbon@silica composite was treated with hydrofluoric acid (10 wt%) at room temperature for 12 h, followed by filtration and washing with ultrapure water (1 L); this procedure was repeated once again to completely remove the silica template. Finally, the powder was washed with ultrapure water (3 L) and then dried under vacuum at 100 °C for 12 h. Caution: after carbonization, a small amount of flammable white phosphorus will be formed on the wall of tube and off-gas pipe, which should be handled carefully. The synthetic procedures of Fe-P$_{700}$-PCC, Fe-P$_{800}$-PCC, Fe-P$_{1000}$-PCC and Fe-P$_{1100}$-PCC are similar to that of Fe-P$_{900}$-PCC, except the final carbonization temperatures were changed to 700, 800, 1000 and 1100 °C, respectively.

**Synthesis of Fe-C$_{900}$-PCC**. 12.5 g sucrose was dissolved into 50 g ultrapure water, followed by addition of 20 g silica colloid, the mixture was stirred at room temperature for 12 h to obtain a homogeneous slurry. The procedures of drying, carbonization, and template removing were similar to that of Fe-P$_{900}$-PCC.

**Synthesis of Fe-N$_{900}$-PCC**. 12.5 g sucrose and 5 g cyanamide were dissolved into 50 g ultrapure water, followed by the addition of 20 g silica colloid; the mixture was stirred at room temperature for 12 h to obtain a homogeneous slurry. The procedures of drying, carbonization, and template removing were similar to that of Fe-P$_{900}$-PCC.

**Synthesis of Fe@Fe-C$_{900}$-PCC**. 12.5 g sucrose was dissolved into 50 g ultrapure water; after that, 130 mg Fe(NO$_3$)$_3$ · 9H$_2$O was added and stirred for 15 min. Followed by the addition of 20 g silica colloid, the mixture was stirred at room temperature for 12 h. The procedures of drying, carbonization, and template removing were similar to that of Fe-P$_{900}$-PCC.

**Synthesis of Fe@Fe-N$_{900}$-PCC**. 12.5 g sucrose and 5 g cyanamide were dissolved into 50 g ultrapure water; after that, 130 mg Fe(NO$_3$)$_3$ · 9H$_2$O was added and stirred for 15 min. Followed by the addition of 20 g silica colloid, the mixture was stirred at room temperature for 12 h. The procedures of drying, carbonization, and template removing were similar to that of Fe-P$_{900}$-PCC.

**Synthesis of Fe$_x$/Fe-P$_{900}$-PCC**. 0.5 g Fe-P$_{900}$-PCC was dispersed in 5 mL ultrapure water, and the suspension was stirred at room temperature for 15 min. Subsequently, aqueous Fe(NO$_3$)$_3$ solution (1 mL) with different content of Fe (NO$_3$)$_3$ · 9H$_2$O (5, 10, 20, 40 mg, respectively) was added dropwise to the suspension. After stirring at room temperature for 12 h, the water of suspension was evaporated at 70 °C. The obtained powder was dried under vacuum at 60 °C for 8 h, and then calcined at 450 °C for 2 h under N$_2$. The accurate Fe loadings are 0.11, 0.2, 0.4, 0.95 wt%, respectively, determined by ICP-MS. The resultant catalysts were denoted as Fe$_{0.11}$/Fe-P$_{900}$-PCC, Fe$_{0.2}$/Fe-P$_{900}$-PCC, Fe$_{0.4}$/Fe-P$_{900}$-PCC, Fe$_{0.95}$/Fe-P$_{900}$-PCC.

**Synthesis of Fe-P$_{900}$-PCC-H**. 1 g Fe-P$_{900}$-PCC was loaded to a quartz crucible in the tube furnace and fluxed with H$_2$-N$_2$ (1:9) for 30 min. The furnace was then heated to 800 °C in the H$_2$-N$_2$ atmosphere at a ramp of 5 °C min$^{-1}$ and was held at 800 °C for 2 h. The obtained catalyst was denoted as Fe-P$_{900}$-PCC-H.

**Synthesis of P$_{900}$-PCC-polymer**. Polytris(4-vinylphenyl)phosphane (1 g) was placed into a tube furnace and then carbonized under the Ar atmosphere. The temperature was controllably ramped at a rate of 2 °C min$^{-1}$ to 600 °C, and subsequently 5 °C min$^{-1}$ to 900 °C, finally maintained at 900 °C for 3 h. After cooling to room temperature, the obtained black powder was treated with hydrofluoric acid (10 wt%) at room temperature for 12 h; this procedure was repeated once again. After filtration, the powder was washed with ultrapure water (3 L) and then dried under vacuum at 100 °C for 12 h.

**Synthesis of Fe-P$_{900}$-PCC-polymer**. Polytris(4-vinylphenyl)phosphane (1 g) was dispersed in 10 mL ultrapure water. Then 25 mg Fe(NO$_3$)$_3$ · 9H$_2$O was added, the mixture was stirred at room temperature for 24 h. After that, the water of suspension was evaporated at 70 °C, and then dried under vacuum at 60 °C for 12 h. The obtained material was placed into a tube furnace and carbonized under Ar atmosphere. The temperature was controllably ramped at a rate of 2 °C min$^{-1}$ to 600 °C, and subsequently 5 °C min$^{-1}$ to 900 °C, finally maintaining at 900 °C for

3 h. After cooling to room temperature, the obtained black powder was treated with hydrofluoric acid (10 wt%) at room temperature for 12 h; this procedure was repeated once again. Finally, the powder was washed with ultrapure water (3 L) and then dried under vacuum at 100 °C for 12 h.

**The recommended synthesis process of Fe-P-C catalyst**. Initially, the phytic acid (Aladdin Industrial Cooperation) containing Fe (0.092 wt%) was used as a precursor to synthesize Fe-P-C catalyst; the Fe content of obtained Fe-P$_{900}$-PCC is relatively low (0.071 wt%). Then, we intended to add different amounts of Fe $(NO_3)_3 \cdot 9H_2O$ (26, 78, 130, 182 mg, theoretical Fe contents of the resultant Fe-P-C catalysts are 0.1, 0.3, 0.5, 0.7 wt%, respectively) into the initial precursors mixture of Fe-P$_{900}$-PCC to synthesize Fe-P-C catalysts with higher Fe content. However, the ICP-MS analyses reveal that the Fe content of obtained catalysts are 0.076, 0.069, 0.085 and 0.073 wt%, respectively, which are close to the Fe content of Fe-P$_{900}$-PCC (0.071 wt%) but lower than the corresponding theoretical value. This is due to most of Fe species have been transformed to metallic Fe aggregates in the process of pyrolysis and then washed off by HF at the step of template removing. These results demonstrate that currently used phytic acid already contains enough amount of Fe for the synthesis of an atomically dispersed Fe-P-C catalyst and the addition of external Fe salt is not so necessary. On the other hand, considering that the phytic acid used by other researchers may have a low content of Fe, the addition of external Fe salt is recommended to guarantee the preparation of a favorable Fe-P-C catalyst.

**Materials characterization**. XRD patterns were collected on a Rigaku SmartLab 3Kw instrument using nickel-filtered Cu Kα radiation (40 kV, 30 mA). Raman spectra excited with a semiconductor (excitation wavelength 532 nm) were collected on Labram HR 800. XPS was operated with a Thermo Scientific NEXSA Instrument. XPS samples were prepared via loading the catalysts onto an adhesive tape. The N$_2$ adsorption/desorption isotherms were measured at 77 K using an ASAP 2020 Micromeritics Instrument. Before analysis, the catalysts were degassed at 180 °C for 4 h. Specific surface areas of catalysts were calculated from the adsorption data obtained at $P/P_0$ between 0.1 and 0.3, using the BET equation. Pore size distributions were obtained from the desorption branch isotherm using the Barrett−Joyner−Halenda method. TEM and HRTEM images of catalysts were obtained with FEI Tecnai G$^2$ F20 S-Twin electron microscope at an acceleration voltage of 200 kV. TEM samples were prepared by placing a drop of catalyst dispersion onto a holey Cu grid. AC-STEM images were obtained on a FEI TITAN Chemi STEM, operating at 200 kV.

The X-ray absorption spectra (XAS) including X-ray absorption near-edge structure (XANES) and extended X-ray absorption fine structure (EXAFS) of the samples were collected at Beamline 17C at the Taiwan Light Source (TLS), Beamline 44A at the Taiwan Photon Source (TPS), and BL14W1 at the Shanghai Synchrotron Radiation Facility (SSRF). All spectra were collected in ambient conditions. Data analysis was carried out with Athena and Artemis included in the Demeter package.

Solid-state $^{31}$P NMR measurements were performed on a Bruker MSL 600 NMR spectrometer with a magnetic field strength of 14.1 T. MAS rotation frequencies is 10 kHz, chemical shift referenced to $(NH_4)_2HPO_4$.

$^{57}$Fe Mössbauer spectrum of Fe-P$_{900}$-PCC was collected and analyzed at the Mössbauer Effect Data Center of the Dalian Institute of Chemical Physics.

The contents of Fe in catalysts were measured with the Thermo Fisher ICAP RQ instrument. Process of sample treatment: 200 mg catalyst was placed in a quartz crucible and calcined at 800 °C for 4 h under the air atmosphere. After cooling to room temperature, 2 mL mixture acid (HCl:HNO$_3$ = 3:1) was added into the quartz crucible and soaked for 12 h. Before running the test, the mixture was filtrated, and the filtrate was diluted by ultrapure water.

Isotopic H-D exchange experiments were performed on the Micromeritics Autochem III unit (Micromeritics Instrument) connected with an OMNI Star mass spectrometer. Two hundred milligrams catalyst was loaded in a U-shaped quartz reactor. The catalyst was first treated at 400 °C for 1 h under He flow (20 mL min$^{-1}$). After cooling to room temperature, the gas flow was changed to H$_2$-D$_2$ mixture gas (H$_2$: 10 mL min$^{-1}$, D$_2$: 15 mL min$^{-1}$) and the baseline was recorded by the mass spectrometer. After a stable baseline was obtained, the temperature of the reactor was increased from room temperature to 600 °C with a heating rate of 3 °C min$^{-1}$.

**Computational methods**. First-principles calculations were performed using periodic density functional (DFT) theory, as implemented in the Vienna Ab initio Simulation Package (VASP)[71,72]. The spin-polarized generalized gradient approximation (GGA) with the PBE functional[73] was used to treat exchange-correlation effects. A plane wave basis set with a cutoff energy of 400 eV was selected to describe the valence electrons. The electron−ion interactions were described by the projector augmented wave (PAW)[74,75] method. Brillouin zone integration was performed with a $3 \times 3 \times 1$ Monkhorst-Pack[76] (MP) k-mesh and Gaussian smearing ($\sigma = 0.1$ eV). We used Grimme's DFT-D3[77] scheme to treat the van der Waals interactions semi-empirically. The self-consistent field (SCF) and force convergence criteria for structural optimization were set to $1 \times 10^{-5}$ eV and 0.01 eV Å$^{-1}$, respectively. The climbing image nudged elastic band (CI-

NEB[78] and dimer methods[79,80] were used to optimize the transition state structures to achieve a force criterion of 0.03 eV Å$^{-1}$. All transition states have been confirmed with the existence of one imaginary frequency whose corresponding eigenvector points in the direction of the reactant and product state. Neighboring slabs were separated by at least 15 Å vacuum. The adsorption energy of a gas-phase molecule is defined as $E_{ads} = E_{(surface + adsorbent)} - E_{(surface)} - E_{(adsorbent)}$.

**Catalytic performance examination**. Substrates, catalyst, and solvent were added into a 30 mL high-pressure autoclave, which was flushed with pure H$_2$ for five times before it was filled with H$_2$ at a certain pressure. When the reaction completed, the reactor was cooled to room temperature and H$_2$ was released. And then, internal standard and 2 mL solvent were added into the autoclave. After centrifugation, the reaction mixture was qualitatively analyzed by gas chromatography in combination with mass spectrometry (GC-MS, Agilent 5975C/7890A equipped with a HP-5MS column), and quantitative analysis was performed on gas chromatography with flame ionization detector (GC-FID, Agilent 6890A) with SE-54 column.

For continuous flow reaction test in fixed-bed reactor, 0.25 g Fe-P$_{900}$-PCC and 0.25 g fumed SiO$_2$ were mixed and formed to 40−100 mesh particles; after that, the mixed catalyst was loaded into the tubular reactor. The reaction was conducted under 4 MPa H$_2$ at 100 °C. H$_2$ and 4-chloronitrobenzene (10 wt% in toluene) were continuously introduced into the reactor at the flow rate of 50 mL min$^{-1}$ and 0.05 mL min$^{-1}$, respectively. The conversion of 4-chloronitrobenzene and yield of 4-chloroaniline were determined by GC-FID.

**Recyclability test**. After each reaction, the catalyst was recovered by centrifugation, and washed with ethanol (10 mL) for three times, dried under vacuum at 60 °C for 4 h. The obtained catalyst was submitted to the next batch of reaction.

## Data availability

All the relevant data are available from the corresponding author upon reasonable request. Source data are provided with this paper.

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

## Acknowledgements

This work was funded by the National Key R&D Program of China (2018YFB1501600), the Natural Science Foundation of China (21972151, 21773271, and 21802149), the Light of West China of the Chinese Academy of Sciences (CAS), the DNL Cooperation Fund of CAS (DNL180303), and the Key Research Program of Frontier Science of CAS (QYZDJSSW-SLH051). We thank Dr. Y. Xi for the help on mechanism study. The $^{57}$Fe Mössbauer spectrum was collected and analyzed at Mössbauer Effect Data Center of the Dalian Institute of Chemical Physics. We also thank beamline 17C at the Taiwan Light Source (TLS), beamline 44A at the Taiwan Photon Source (TPS) and beamline BL14W1 at the Shanghai Synchrotron Radiation Facility (SSRF).

## Author contributions

X.L. and G.G. conducted the experiments, analyzed the results and wrote the manuscript. F.L. designed the research, supervised the project, and edited the manuscript. Z.L., P.S. and J.W. participated in writing the manuscript. B.Z. performed the AC-STEM characterizations. J.Z. and Z.J. performed the XAS experiments and conducted the calculations.

## Competing interests

The authors declare no competing interests.
