## [Peer Review File · Nature Communications]

Reviewers' Comments:

Reviewer #1:

Remarks to the Author:

This work report atomically dispersed metal-phosphorus-carbon (M-P-C) catalysts exhibited outstanding catalytic performance and reaction generality in a series heterogeneous hydrogenation reaction. Considering the wide applications of transition metal phosphine complexes in various homogeneous catalytic transformation reactions, the outstanding catalytic activity and recyclability of the catalysts demonstrated in this work inspired the development of various metal-phosphorus-carbon (M-P-C) recyclable catalysts for a wide range of heterogeneous reactions. I suggested this work publish in Nature communication with the following comments.

1. The work used Fe species in the raw phytic acid as the Fe resource. What is the concentration of Fe content in the raw phytic acid resource?
2. This work demonstrated that the analogues of single metal-nitrogen-carbon (M-N-C) catalysts were not active for these reactions, clearly demonstrating the unique coordinated environment provided by P compared to that of commonly used N. A hypothesis should be provided for this interesting phenomena. A molecular leveled catalytic mechanism could further enhance the scientific impact of this work.
3. It was also demonstrated that the existence of both Fe and graphitic P in the catalysts are critical. Even though multiple techniques already used to characterize the catalysts, it is still not clear the structure of the catalytic center. For example, how many P atoms, C or even O are coordinated with the Fe?

Reviewer #2:

Remarks to the Author:

In this work, the authors have reported the synthesis of atomically dispersed metal-phosphorus-carbon (M-P-C) catalyst with graphitic phosphorus species (P_{grap}) coordinated single-atom Fe on P-doped carbon, displaying enhanced catalytic performance towards the heterogeneous hydrogenation reactions relative to the inactive atomically dispersed Fe atoms embedded on N-doped carbon. Although the manuscript is well written, the novelty of synthesis strategy is insufficient, the experimental control and repeatability is suspicious to some degree, and the advantage of Fe-P-C catalysts compared with other common catalysts is inaccessible. Specific commitments as follows should be addressed in any possible revision, and then further consideration can be made.

1. Directly utilizing raw materials contained metal as active components to construct Fe-P-C catalysts is not a very convincing routes, since the impurity is usually ambiguous. Fe-P900-PCC showed Fe³⁺, is it different from Fe³⁺ precursor salt? If so, is it possible to synthesize Fe-P900-PCC by adding Fe³⁺ precursor salt? The authors should design a more controllable synthesis process, just like the conditions for Fe@Fe-C900-PCC. However, no detailed characterizations besides ICP-MS are found for Fe@Fe-C900-PCC catalysts, such as AC-HTEM, XPS, EXAFS.
2. Why the different annealing temperature leads to different Fe content?
3. Why there are obvious difference in the surface area for Fe-C900-PCC, Fe-N900-PCC, Fe-P900-PCC samples? Is there any randomness in the controlling of pore structure or heteroatom effects?
4. A high density of single Fe atoms in Fe-N900-PCC, Fe-P900-PCC with a very low Fe content may be conflicting. EELS analysis should be provided to confirm the local coordination environment.
5. The graphitic P group is different from the Fe-P. Therefore, the active center for hydrogenation reactions is the graphitic P group, or Fe-P, or graphitic P group-Fe? It should be clear. Moreover, the fitted P XPS spectra showed that no Fe-P state while XAFS results presented Fe-P state, it is contradictory. And the content of P in various samples should be measured by ICP, because of the incorrect content measured by XPS. It is complex that the content of P presented atomic % and the content of Fe present wt.%.
6. If the chemical state of Fe-P900-PCC is significant for the hydrogenation, the chemical state of the spent catalyst after the stability test is necessary.
7. A comprehensive comparison with relative reports should be in Tables to demonstrate the catalytic performance.
8. Conclusions should not be involved in discussion and theoretical calculation of active sites is suggested here.

Reviewer #3:

Remarks to the Author:

In this manuscript, the authors reported that graphitic phosphorus species (P_{grap}) coordinated single-atom Fe on are active in hydrogenation of N-heterocycles, functionalized nitroarenes, and reductive amination reactions, while single-atom Fe atoms on either carbon or N-doped carbon are almost inactive. XPS was employed to verify the correlations between the graphitic P and hydrogenation activity. XAFS and STEM confirmed the atomic presence of Fe. The manuscript might be publishable in this high standard journal after addressing the following questions:

1. One of the major drawbacks of this manuscript is lack of the atomic structure of this Fe-P-C catalyst. What are the coordination numbers of Fe-P? How Fe is coordinated with graphitic P? What is the structure?
2. The authors might also add some additional discussion about the hydrogenation mechanism.
3. On page 6, the authors claimed that "the formation of P_{grap} is the prerequisite for the existence of single Fe atoms on the surface of P-doped porous carbon". What is the evidence for this? Then what is the form of the Fe in Fe-P₇₀₀-PCC?
4. On page 5, the authors said that "even though the AC-STEM has observed the presence of some Fe nanoparticles in Fe-P₉₀₀-PCC (Supplementary Fig. 10), these nanoparticles are deeply embedded in the carbon matrix and could not be detected by the high-energy X-ray. A" This is not correct, since XAFS is a bulk technique.
5. AC-STEM images of Fe-C₉₀₀-PCC are missing. They be useful to confirm the presence of Fe atoms in this sample.
6. On page 8, it should be "summary or conclusion" rather than "discussion".
7. This reviewer is wondering what are the activities of Fe nanoparticles on carbon or N-doped carbon supports, it might be useful to make the comparison as well.

Responses to Reviewers' Comments

Reviewer #1

Comment. *This work report atomically dispersed metal-phosphorus-carbon (M-P-C) catalysts exhibited outstanding catalytic performance and reaction generality in a series heterogeneous hydrogenation reaction. Considering the wide applications of transition metal phosphine complexes in various homogeneous catalytic transformation reactions, the outstanding catalytic activity and recyclability of the catalysts demonstrated in this work inspired the development of various metal-phosphorus-carbon (M-P-C) recyclable catalysts for a wide range of heterogeneous reactions. I suggested this work publish in Nature communications with the following comments.*

Response. We appreciate reviewer very much for the positive comment, and we are delighted to find that reviewer agrees with the significance of M-P-C single atom catalyst in catalysis.

Comment 1. *The work used Fe species in the raw phytic acid as the Fe resource. What is the concentration of Fe content in the raw phytic acid resource?*

Response 1. Thank you very much for this insightful comment. The Fe contents of all the raw materials for the preparation of porous carbon catalysts were characterized by ICP-MS, and the results are listed in below Supplementary Table 1.

Supplementary Table 1. The contents of Fe in the raw materials.

Raw material	Fe content (wt%)
Silica colloid (Alfa Aesar)	3.5×10^{-4}
Sucrose (Macklin Biochemical Co. Ltd)	9.3×10^{-4}
Phytic acid solution (Aladdin Industrial Cooperation)	9.2×10^{-2}
Cyanamide (Alfa Aesar)	1.5×10^{-3}

Comment 2. *This work demonstrated that the analogues of single metal-nitrogen-carbon (M-N-C) catalysts were not active for these reactions, clearly demonstrating the unique coordinated environment provided by P compared to that of commonly used N. A hypothesis should be provided for this interesting phenomenon. A molecular leveled catalytic mechanism could further enhance the scientific impact of this work.*

Response 2. Thank you very much for this helpful comment. Following your suggestion, the coordination structures of Fe centers in N-doped carbon and P-doped carbon have been investigated by EXAFS curve fitting and theoretical calculations. Meanwhile, the H₂ absorption/activation on single atom Fe-N/P-C catalysts and the hydrogenation process of quinoline over Fe-P-C catalyst were studied by density functional theory (DFT) calculations.

Take Fe@Fe-N₉₀₀-PCC as an example of Fe-N-C catalysts, both its AC-STEM image and Fe k-edge EXAFS spectrum indicate that the Fe atoms are atomically dispersed on the surface of N-doped porous carbon (Supplementary Fig. 12). The EXAFS curve shows a strong Fe-N peak at 1.45 Å, so the quantitative EXAFS fitting was performed by including a single Fe-N shell within the R-range of 1-3.1 Å, which reveals that the coordination number of Fe center with surrounding N atoms is 4.2 ± 0.4 and the average Fe-N bond length is $1.95 \text{ \AA} \pm 0.01$ (Supplementary Table 7), suggesting the single Fe atom in Fe@Fe-N₉₀₀-PCC adopts a planar Fe-N₄ structure as presented in Supplementary Fig. 13b.

For the representative Fe-P₉₀₀-PCC catalyst, its EXAFS spectrum shows that the main peak locates at 1.63 Å, ascribing to Fe-P first shell coordination. Furthermore, the Fe-O first shell coordination at 1.45 Å is also included in this broadening peak, which indicates that O need to be included in the curve fitting. On the other hand, a shoulder peak at 2.55 Å for Fe-C second shell coordination is also observed. Therefore, a three-shell structure model, including a Fe-P, a Fe-O and a Fe-C shell, was initially used to fit the EXAFS data of Fe-P₉₀₀-PCC. The best-fitting analysis manifests that the dominant contribution is originated from Fe-P and Fe-O first shell coordination as presented in Fig. 3c and 3d. The coordination numbers for P and O atoms are calculated as 4.0 ± 0.8 and 2.0 ± 0.4 , and the corresponding mean bond length of Fe-P and Fe-O are $2.35 \text{ \AA} \pm 0.02$ and $2.00 \text{ \AA} \pm 0.03$, respectively. These results reveal that the single Fe atom in Fe-P₉₀₀-PCC coordinates with four P atoms and a dioxygen molecule (O₂-Fe-P₄). Because the atomic size of P (106 pm) is larger than C (75 pm), Fe center adopts a pyramidal geometry as shown in Fig. 3e, this structure is quite different from the planar structure of Fe-N₄. Besides, the theoretical calculations reveal that the binding energy of O₂-Fe-P₄ structure is -2.09 eV, suggesting this coordination structure is thermodynamically stable.

After studying the structural information of the catalytic Fe sites in Fe-N-C and Fe-P-C catalysts, we then performed the DFT calculations to investigate the activity of Fe-N₄ and O₂-Fe-P₄ toward H₂ adsorption/activation. As shown in Supplementary Fig. 14, the DFT calculation reveals that H₂ can only physically adsorb on the Fe-N₄ site, the adsorbed H-H bond length (0.75 Å) is equal to the free H₂ molecule, indicating that the atomically dispersed Fe-N₄ site is intrinsically inert to activate H₂ molecule. This is consistent with the experimental results that both Fe-N₉₀₀-PCC and Fe@Fe-N₉₀₀-PCC dominated by single Fe-N₄ sites were inactive for hydrogenation of quinoline. Interestingly, the DFT calculations demonstrate that the original O₂-Fe-P₄ sites on Fe-P₉₀₀-PCC can be reduced to the Fe-P₄ sites in the presence of H₂ (the reaction of O₂-Fe-P₄ + 2H₂ → Fe-P₄ + 2H₂O is exothermic by 4.08 eV), the H₂ then adsorbs on Fe atom and the H-H bond length is elongated to 0.81 Å with a free

energy of -0.407 eV, suggesting that the H₂ can be activated on the Fe-P₄ sites originated from the present M-P-C catalyst. These results clearly demonstrate the unique coordination environment enabled by P compared to that of commonly used N in the hydrogenation reactions. Finally, we also conducted DFT calculations to understand the mechanism of the hydrogenation reaction of quinoline over Fe-P₉₀₀-PCC (Fig. 4).

Accordingly, we have revised our manuscript in Page 4 as the following:

“For Fe@Fe-N₉₀₀-PCC, the characterizations of AC-STEM and EXAFS demonstrate that the Fe atoms are atomically dispersed on N-doped porous carbon (Supplementary Fig. 12), and the quantitative EXAFS curve fitting reveals that the single Fe atom adopts a planar Fe-N₄ structure (see Supplementary Fig. 13 and Supplementary Table 7 for details). Surprisingly, Fe@Fe-N₉₀₀-PCC has no catalytic activity for hydrogenation (Table 1, entry 9), even though its Fe content is similar to the P-doped Fe-P₉₀₀-PCC. We then performed the density functional theory (DFT) calculations to investigate the catalytic behavior of Fe-N₄ sites toward H₂ adsorption and activation, the results demonstrate that H₂ can only physically adsorb on the Fe-N₄ site, and the resultant H-H bond length (0.75 Å) is equal to the free H₂ molecule (Supplementary Fig. 14), indicating that the Fe-N₄ sites are intrinsically inert to activate H₂, which is consistent with the experimental results.”

The above mentioned Supplementary Fig. 12-14 and Supplementary Table 7 have been added into the revised supplementary information as below:

Supplementary Fig. 12 Characterizations of Fe@Fe-N₉₀₀-PCC: (a) AC-STEM image, Fe single atoms are highlighted by yellow circles, and (b) EXAFS spectra of Fe-N₉₀₀-PCC and Fe foil.

Supplementary Fig. 13 The Fourier-transform experimental Fe k-edge EXAFS spectrum (blue dots) and the fitted curve (red line) of Fe@Fe-N₉₀₀-PCC: **(a)** *k*-space, **(b)** R-space. The inset in **(b)** demonstrates the schematic model (Fe-N₄) derived from the EXAFS results, C (●), N (●), Fe (●).

Supplementary Fig. 14 The top **(left)** and side **(right)** views of optimized adsorption structures of H₂ molecule on Fe-N₄. The H-H bond length of adsorbed H₂ molecule is 0.75 Å, indicating that the Fe-N₄ site is incapable of activating H₂ molecule. Color code: C (●), H (●), N (●), Fe (●).

Supplementary Table 7. Fitting results of Fe k-edge EXAFS data for Fe@Fe-N₉₀₀-PCC^a and Fe-P₉₀₀-PCC^b.

Sample	Bond	CN	R (Å)	σ^2 (Å ²)	ΔE_0 (eV)	R-factor
Fe@Fe-N ₉₀₀ -PCC	Fe-N	4.2 ± 0.4	1.95 ± 0.01	0.007 ± 0.001	-6.23 ± 1.1	0.004
Fe-P ₉₀₀ -PCC	Fe-P	4.0 ± 0.8	2.35 ± 0.02	0.014 ± 0.006	-0.96 ± 0.3	0.013
	Fe-O	2.0 ± 0.4	2.00 ± 0.03	0.004 ± 0.004	-0.96 ± 0.3	

The average lengths of Fe-N, Fe-P and Fe-O bonds and coordination numbers of Fe atoms are extracted from the curve fitting for Fe k-edge EXAFS data (Supplementary Fig. 13 and Manuscript Fig. 3). CN, coordination number; R, distance between absorber and backscatter atoms; σ^2 , the Debye-Waller factor; ΔE_0 , inner potential correction; R-factor, indicate the goodness of the fit.

^aFor the EXAFS spectrum of Fe@Fe-N₉₀₀-PCC (Supplementary Fig. 12), only a strong Fe-N peak at 1.45 Å is observed. So, the fitting was performed by including a single Fe-N shell within the R-rang of 1 - 3.1 Å and *k*-rang of 1.42 Å⁻¹ - 9.62 Å⁻¹. The fitting results reveal that the coordination number of Fe center with surrounding N atoms is 4.2 ± 0.4 and the average Fe-N bond length is 1.95 ± 0.01 Å, suggesting the single Fe sites in Fe@Fe-N₉₀₀-PCC adopt a planar Fe-N₄ structure (as presented in Supplementary Fig. 13b).

^bThe EXAFS spectrum of Fe-P₉₀₀-PCC shows that the main peak locates at 1.63 Å, ascribing to Fe-P first shell coordination. Furthermore, the Fe-O first shell coordination at 1.45 Å is also included in this broadening peak, which indicates that O need to be included in the curve fitting. On the other hand, a shoulder peak at 2.55 Å for Fe-C second shell coordination is also observed. Therefore, a three-shell structure model, including a Fe-P, a Fe-O and a Fe-C shell, is initially used to fit the EXAFS data of Fe-P₉₀₀-PCC within the R-rang of 1 - 3.1 Å and *k*-rang of 1.42 Å⁻¹ - 9.62 Å⁻¹. The best-fitting analyses manifests that the dominant contribution is originated from Fe-P and Fe-O first shell coordination as presented in Manuscript Fig. 3c and 3d. The coordination numbers for P and O atoms are calculated as 4.0 ± 0.8 and 2.0 ± 0.4, and the corresponding mean bond length of Fe-P and Fe-O are 2.35 ± 0.02 Å and 2.00 ± 0.03 Å, respectively. These results reveal that the single Fe atom in Fe-P₉₀₀-PCC coordinates with four P atoms and a dioxygen molecule (O₂-Fe-P₄). Because the atomic size of P (106 pm) is larger than C (75 pm), Fe center adopts a pyramidal geometry as shown in Manuscript Fig. 3e, this structure is quite different from the planar structure of Fe-N₄.

Subsequently, we have revised our manuscript in Page 6 as the following:

“The coordination information of Fe atom in optimal Fe-P₉₀₀-PCC is further studied using quantitative EXAFS curve fitting (see Supplementary Table 7 for details). The best-fitting analyses are shown in Fig. 3c and 3d, which manifests that the Fe atom are coordinated by four P atoms and a dioxygen molecule (O₂-Fe-P₄), forming a pyramidal geometry as shown in Fig. 3e, this structure is quite different from the planar structure of Fe-N₄. Besides, the theoretical calculations reveal that the binding energy of O₂-Fe-P₄ structure is -2.09 eV, suggesting this coordination structure is thermodynamically stable.”

The revised Fig. 3 has been added into page 6 of the revised manuscript as below:

Fig. 3 Characterizations of Fe species. **a** EXAFS spectra of Fe-P_x-PCCs, Fe_{0.4}/Fe-P₉₀₀-PCC, and reference materials (Fe foil and Fe₂O₃). **b** ⁵⁷Fe Mössbauer spectrum of Fe-P₉₀₀-PCC. **c** and **d** The experimental Fe k-edge EXAFS spectrum (blue dot) and the fitting curve (red line) of Fe-P₉₀₀-PCC at *k* and *R* space, respectively. **e** Top and side views of optimized structural model of O₂-Fe-P₄.

Finally, we have revised our manuscript in Page 7 as below:

“DFT calculations. To understand the possible mechanism for the hydrogenation of quinoline, we studied the reaction process with DFT calculations, which initially reveals that the bond length of O-O in O_2 -Fe-P₄ is 1.314 Å, the two oxygen atoms have a total magnetic moment of 0.77 uB, O_2 gains approximately one electron from Fe-P₄ upon adsorption, and the adsorbed O_2 is assigned as O_2^- which can be readily reduced by H_2 since the reaction (O_2 -Fe-P₄ + $2H_2 \rightarrow$ Fe-P₄ + $2H_2O$) is exothermic by 4.08 eV. Therefore, the Fe-P₄ was used as the starting point to calculate the reaction pathway of hydrogenation of quinoline. As shown in Fig. 4, the H_2 preferentially adsorbs on Fe atom, the bond length of H-H is elongated to 0.81 Å with a free energy of -0.407 eV, indicating that the H_2 can easily adsorb and bond on Fe-P₄. On the other hand, the adsorption energy of quinoline on Fe atoms is -0.687 eV (Supplementary Table 9 and 10). Then, one hydrogen is transferred to N atom ($C_9H_8N^*$), leaving another hydrogen bound to Fe atom (int-2 to int-3). Hydrogenation of $C_9H_8N^*$ to adsorbed $C_9H_9N^*$ proceeds with an energy barrier of 0.38 eV and is exothermic by -0.348 eV (from int-3 to int-4). The reaction continues to give final product via the addition of second hydrogen molecule, and the reaction of $C_9H_9N^*$ to $C_9H_{10}N^*$ (int-5 to int-6) is found to be the rate determining step with an energy barrier of 0.728 eV.”

The above mentioned Fig. 4 is added into the revised manuscript as below:

Fig. 4 Energy profile of hydrogenation of quinoline over Fe-P₉₀₀-PCC.

Comment 3. *It was also demonstrated that the existence of both Fe and graphitic P in the catalysts are critical. Even though multiple techniques already used to characterize the catalysts, it is still not clear the structure of the catalytic center. For example, how many P atoms, C or even O are coordinated with the Fe?*

Response 3. Thank you very much for this insightful comment. As discussed above, the EXAFS curve fitting and theoretical calculations revealed that the Fe center adopts a pyramidal O₂-Fe-P₄ structure as shown in Fig. 3e, and the corresponding DFT calculations indicate that the O₂-Fe-P₄ structure can be reduced to the active Fe-P₄ sites, which can further catalyze the activation of H₂ and the hydrogenation as shown in Fig. 4.

Reviewer #2

Comment. *In this work, the authors have reported the synthesis of atomically dispersed metal-phosphorus-carbon (M-P-C) catalyst with graphitic phosphorus species (P_{grap}) coordinated single-atom Fe on P-doped carbon, displaying enhanced catalytic performance towards the heterogeneous hydrogenation reactions relative to the inactive atomically dispersed Fe atoms embedded on N-doped carbon. Although the manuscript is well written, the novelty of synthesis strategy is insufficient, the experimental control and repeatability is suspicious to some degree, and the advantage of Fe-P-C catalysts compared with other common catalysts is inaccessible. Specific commitments as follows should be addressed in any possible revision, and then further consideration can be made.*

Response. Thank you very much for this insightful comment. We agree with the reviewer that the pyrolysis method is not a novel strategy to fabricate single atom catalysts, although it is a useful and convenient button-up method to synthesize carbon-based single atom catalysts by using small molecule as precursor¹⁻⁴. In this area, the most intensively studied material are Fe-N-C single atom catalysts, these catalysts were mainly applied in different electrocatalytic reactions and oxidation reactions under thermal condition. Notably, the atomically dispersed metal-phosphorus-carbon (M-P-C) catalyst has not yet been fabricated. Follow your helpful suggestions, we have added more experimental fabrications and revised the catalyst synthetic procedure to be clearer and repeatable. In addition, we also compared the catalytic activities of our Fe-P-C catalyst with the related catalysts in specific hydrogenation reactions.

References

1. Wang, G. *et al.* *J. Am. Chem. Soc.* **141**, 20118-20126 (2019).
2. Zhang, T. *et al.* *J. Am. Chem. Soc.* **139**, 10790-10798 (2017).
3. Scherf, U. *et al.* *J. Am. Chem. Soc.* **142**, 2404-2412 (2020).
4. Hu, X. *et al.* *Science.* **363**, 1091-1094 (2019).

Comment 1. *Directly utilizing raw materials contained metal as active components to construct Fe-P-C catalysts is not a very convincing routes, since the impurity is usually ambiguous. Fe-P₉₀₀-PCC showed Fe³⁺, is it different from Fe³⁺ precursor salt? If so, is it possible to synthesize Fe-P₉₀₀-PCC by adding Fe³⁺ precursor salt? The authors should design a more controllable synthesis process, just like the conditions for Fe@Fe-C₉₀₀-PCC.*

Response 1. Thank you very much for this very important comment. Exactly as the reviewer mentioned that the main problem of using raw materials contained metal as active component to synthesize catalyst is the metal contents of raw material and resultant catalyst are uncontrollable. In

this work, the Fe content of phytic acid is 0.092 wt%, and the obtained Fe-P₉₀₀-PCC has a content of Fe (0.071 wt%). In order to investigate whether the addition of external Fe salt could increase the Fe content of catalyst, a series of different amounts of Fe(NO₃)₃·9H₂O (26 mg, 78 mg, 130 mg, 182 mg, theoretical Fe contents of the resultant Fe-P-C catalysts are 0.1 wt%, 0.3 wt%, 0.5 wt%, 0.7wt%, respectively) have been added into the precursors mixture of Fe-P₉₀₀-PCC. The obtained catalysts are denoted as Fe_x@Fe-P₉₀₀-PCC, wherein x represent the theoretical weight percentage of Fe. The ICP-MS analysis reveal that the Fe contents of Fe_{0.1}@Fe-P₉₀₀-PCC, Fe_{0.3}@Fe-P₉₀₀-PCC, Fe_{0.5}@Fe-P₉₀₀-PCC and Fe_{0.7}@Fe-P₉₀₀-PCC are 0.076, 0.069, 0.085 and 0.073 wt%, respectively, which are close to the Fe content of Fe-P₉₀₀-PCC (0.071 wt%) but is lower than the corresponding theoretical value. The main reason why the experimental Fe content is much lower than the theoretical value is that most of Fe species have been transformed to metallic Fe aggregates in the process of pyrolysis and then washed off by HF at the step of template removing. Meanwhile, these results demonstrated that currently used phytic acid already contains enough amount of Fe for the synthesis of an atomically dispersed Fe-P-C catalyst and the addition of external Fe is not so necessary. When Fe_{0.1}@Fe-P₉₀₀-PCC, Fe_{0.3}@Fe-P₉₀₀-PCC, Fe_{0.5}@Fe-P₉₀₀-PCC and Fe_{0.7}@Fe-P₉₀₀-PCC are used as catalysts for hydrogenation of quinoline, the conversion of quinoline (93%, 92%, 93% and 91%, respectively) are equal to that of Fe-P₉₀₀-PCC (93%). However, considering that the phytic acid used by other researchers may have a low content of Fe, addition of external Fe salt is recommended to guarantee the preparation of a favorable Fe-P-C catalyst.

To ensure the present Fe-P₉₀₀-PCC single atom catalyst can be reproduced by others, a recommended synthesis process has been added into the Supplementary Information (Page 4) as below:

“Recommended synthesis process of Fe-P-C single atom catalyst:

Initially, the phytic acid (Aladdin Industrial Cooperation) contained Fe (0.092 wt%) was used as active component to synthesize Fe-P-C catalyst, the Fe content of obtained Fe-P₉₀₀-PCC is relatively low (0.071 wt%). We have intended to add different amounts of Fe(NO₃)₃·9H₂O (26 mg, 78 mg, 130 mg, 182 mg, theoretical Fe contents of the resultant Fe-P-C catalysts are 0.1 wt%, 0.3 wt%, 0.5 wt%, 0.7wt%, respectively) into the initial precursors mixture of Fe-P₉₀₀-PCC to synthesize Fe-P-C catalysts with higher Fe content. However, the ICP-MS analyses reveal that the Fe content of obtained catalysts are 0.076, 0.069, 0.085 and 0.073 wt%, respectively, which are close to the Fe content of Fe-P₉₀₀-PCC (0.071 wt%) but lower than the corresponding theoretical value. This is due to most of Fe species have been transformed to metallic Fe aggregates in the process of pyrolysis and then washed off by HF at the step of template removing. These results demonstrated that currently used phytic acid already contains enough amount of Fe for tht synthesis an atomically dispersed Fe-P-C catalyst and the

addition of external Fe salt is not so necessary. On the other hand, considering that the phytic acid used by other researchers may have a low content of Fe, addition of external Fe salt is recommended to guarantee the preparation of a favorable Fe-P-C catalyst.”

Comment 2. *However, no detailed characterizations besides ICP-MS are found for Fe@Fe-C₉₀₀-PCC catalysts, such as AC-HRTEM, XPS, EXAFS.*

Response 2. Following your suggestion, AC-STEM images of Fe@Fe-C₉₀₀-PCC were acquired as displayed in Supplementary Fig. 11a and 11b, which indicate that there are some of single Fe atoms dispersed on the surface of Fe@Fe-C₉₀₀-PCC, and Fe nanoparticles are also observed in this catalyst (Supplementary Fig. 11c). Furthermore, the corresponding Fe k-edge EXAFS spectrum (Supplementary Fig. 11d) also reveals that both Fe single atoms and nanoparticles are dispersed on Fe@Fe-C₉₀₀-PCC, since the main shell peak at 1.45 Å ascribed to Fe-C coordination and a small peak of Fe-Fe coordination at 2.19 Å are observed. The Fe 2p XPS spectrum of Fe@Fe-C₉₀₀-PCC reveal that the Fe species is in the +3 oxidation state¹ (Figure R1).

Reference

1. Sullivan, J. *et al. J. Phys. D Appl. Phys.* **16**, 723-732 (1983)

Revisions made in Page 4 to describe the surface property of Fe@Fe-C₉₀₀-PCC catalyst:

“AC-STEM and HRTEM images as well as Fe k-edge EXAFS spectrum reveal that both Fe single atoms and nanoparticles are dispersed on Fe@Fe-C₉₀₀-PCC (Supplementary Fig. 11).”

Supplementary Fig. 11 has been added into Supplementary Information:

Supplementary Fig. 11 Characterizations of Fe@Fe-C₉₀₀-PCC: (a) and (b) AC-STEM images, Fe single atoms are highlighted by yellow circles, (c) HRTEM image, and (d) EXAFS spectra of Fe-N₉₀₀-PCC and reference materials (Fe foil and Fe₂O₃).

Figure R1. XPS Fe 2p spectrum of Fe@Fe-C₉₀₀-PCC.

Comment 3. *Why the different annealing temperature leads to different Fe content?*

Response 3. Thank you very much for this insightful comment. In this work, the porous carbon catalysts were synthesized by a hard-template method. After precursors were pyrolyzed under inert atmosphere, the resultant Fe@carbon@SiO₂ composite need to be washed two times by hydrofluoric acid to remove SiO₂ template. Meanwhile, most of metallic Fe nanoparticles formed in the process of

carbonization were also removed by hydrofluoric acid. Therefore, the remained Fe species in the finally obtained catalysts mainly are graphitic phosphorus coordinated single Fe atoms and a certain amount of carbon layers buried Fe nanoparticles. It should be pointed out that the Fe contents of Fe-P_x-PCCs (x stands for annealing temperature, 700 to 1100 °C) were characterized by a “combustion method”, wherein the catalysts were firstly calcined under air atmosphere, then the obtained residual were dissolved by aqua regia and tested by ICP-MS. In this way, both of graphitic P coordinated single Fe atoms and carbon layers buried Fe nanoparticles are included in the total Fe content. It has been found in the fabrication of Fe-P_x-PCCs that increasing the annealing temperature from 700 to 900 °C leads to more insertion of P atoms into the skeleton of graphite to form graphitic P, which could coordinate and stabilize with more Fe atoms, and thus the Fe content of Fe-P₇₀₀-PCC, Fe-P₈₀₀-PCC and Fe-P₉₀₀-PCC increased gradually. After raising the annealing temperature to 1000 and 1100 °C, although the contents of graphitic P and the coordinated single Fe atoms are reduced, these higher annealing temperatures are greatly beneficial to the formation of carbon layers buried Fe nanoparticles¹, which is the reason why the Fe contents of Fe-P₁₀₀₀-PCC and Fe-P₁₁₀₀-PCC are higher than Fe-P₉₀₀-PCC.

Reference

1. Pérez-Ramírez, J. *Angew. Chem. Int. Ed.* **58**, 12297-12304 (2019).

Comment 4. *Why there are obvious difference in the surface area for Fe-C₉₀₀-PCC, Fe-N₉₀₀-PCC, Fe-P₉₀₀-PCC samples? Is there any randomness in the controlling of pore structure or heteroatom effects?*

Response 4. Thank you so much for your valuable comment. As shown in Supplementary Fig. 2, Fe-C₉₀₀-PCC and Fe-N₉₀₀-PCC have the similar N₂ adsorption-desorption isotherms and pore size distributions, indicating that these two materials have similar porous structure. However, the XRD patterns show that there is a difference in the diffraction of (002) (Supplementary Fig. 3). The (002) diffraction of Fe-C₉₀₀-PCC locates at 23.1°, while the (002) diffraction of Fe-N₉₀₀-PCC shifts to 25.4°, suggesting the decrease of interplanar spacing after N doping into the carbon framework. Based on the Scherrer equation (1) and (2), it can be concluded that the enlarged diffraction angle will leads to the decrease of surface area. Therefore, the heteroatom doping effect is the reason for decrease of the surface area of Fe-N₉₀₀-PCC when compare with Fe-C₉₀₀-PCC¹.

$$D = k\lambda / \beta \cos\theta \quad (1)$$

$$S = 6 / \rho D \quad (2)$$

D is the mean size of the crystalline; k is the Scherrer constant (0.89); λ is the X-ray wavelength; β is the peak width half-height; θ is the Bragg angle; ρ is the crystal density; S is the surface area.

For Fe-P₉₀₀-PCC, the phytic acid was used as phosphorus precursor. In fact, it is also a relatively strong acidic reagent, which will promote the thermal polymerization of carbon precursor (sucrose) and improve the condensation degree of carbon material, just like the role of H₂SO₄ used in the preparation process of CMK-3². Therefore, it is speculated that the acidity of phytic acid results in the larger pore size and lower specific surface area of Fe-P₉₀₀-PCC compared with corresponding Fe-C₉₀₀-PCC and Fe-N₉₀₀-PCC (Supplementary Fig. 2b).

References

1. Shakir, I. *et al. Ceram. Int.* **46**, 12884-12890 (2020).
2. Ryong, R. *et al. J. Am. Chem. Soc.* **122**, 10712-10713 (2000).

Supplementary Fig. 2 (a) N₂ adsorption-desorption isotherms and (b) pore size distributions of Fe-C₉₀₀-PCC, Fe-N₉₀₀-PCC, and Fe-P₉₀₀-PCC.

Supplementary Fig. 3 XRD patterns of Fe-C₉₀₀-PCC, Fe-N₉₀₀-PCC, and Fe-P₉₀₀-PCC.

Comment 5. A high density of single Fe atoms in Fe-N₉₀₀-PCC, Fe-P₉₀₀-PCC with a very low Fe content may be conflicting. EELS analysis should be provided to confirm the local coordination environment.

Response 5. Thank you very much for the nice suggestion. We are sorry for this conflicting statement caused by the improper use of language and we have corrected this improper expression as the following (Page 3):

“As shown in Fig. 1d, Supplementary Fig. 7 and 8, a high density of single Fe atoms is uniformly dispersed on both Fe-P₉₀₀-PCC and Fe-N₉₀₀-PCC.” has been changed to “As shown in Fig. 1d, Supplementary Fig. 7, and 8, single Fe atoms are dispersed on these catalysts.”

EELS is a powerful analytical technique to determine the local coordination state of single metal atom. As your suggestion, we have performed EELS characterization to probe the local coordination information of Fe atoms in Fe-P₉₀₀-PCC. However, because the detection limit of TEM’s energy spectrum we utilized is relatively low, the EELS atomic spectrum of Fe element is not pronounced and cannot gain useful information (Figure R2).

Figure R2. The EELS atomic spectra of P, O and Fe elements in Fe-P₉₀₀.PCC.

Comment 6. *The graphitic P group is different from the Fe-P. Therefore, the active center for hydrogenation reactions is the graphitic P group, or Fe-P, or graphitic P group-Fe? It should be clear.*

Response 6. We appreciate the reviewer very much for this insightful comment, which prompts us to explore more structural information about the active sites in Fe-P-C catalyst.

Actually, identifying the active site is one of the main points of this work, and most of experiments and characterizations are around this theme. The distinctive catalytic performance of Fe-P₉₀₀-PCC in hydrogenation provokes us to discriminate the active sites in this catalyst. Initially, we deduced that the sole Fe species are not the active sites, because Fe-C₉₀₀-PCC and Fe@Fe-C₉₀₀-PCC were inert in hydrogenation of quinoline. It was then proposed that the functional P groups on the surface of Fe-P₉₀₀-PCC had the hydrogenation activity, since previous works have reported that the metal-free carbon material or N/P doped carbon materials can catalyze the hydrogenation reaction¹⁻³. We subsequently synthesized an ultrapure P-doped carbon (denoted as P₉₀₀-PCC-polymer, only 0.000023

wt% of Fe) via pyrolysis of P containing organic polymer. The gas phase isotopic H-D exchange experiment over P₉₀₀-PCC-polymer demonstrated that this P-doped carbon cannot activate the H₂ (Supplementary Fig. 15). On the other hand, we synthesized a control Fe catalyst, Fe-P₉₀₀-PCC-polymer, through pyrolysis of Fe coordinated P containing organic polymer, and then the same gas phase H-D exchange experiment revealed that the H₂ can be effectively activated over this catalyst (Supplementary Fig. 15). These two control experiments indicate that the P functional groups cannot catalyze the hydrogenation reaction, and it was speculated that the Fe species interacted with the P species in Fe-P₉₀₀-PCC co-catalyzed the H₂ dissociation and hydrogenation reaction.

The P 2p XPS spectrum of Fe-P₉₀₀-PCC demonstrates that there are three kinds of P functional groups on the surface of Fe-P₉₀₀-PCC: C-O-P, C-PO₂/C₂-PO₂ and graphitic P. To determine which P group coordinated Fe is the active sites, we then fabricated a series of Fe-P_x-PCC (x stands for carbonization temperature, 700-1100 °C). It was revealed that the content of graphitic P is positively associated with the catalytic activities, which might mean graphitic P coordinated Fe is the active sites for hydrogenation reaction, rather than graphitic P itself, because above H-D exchange experiment has approved the P-doped carbon catalyst cannot disassociate the H₂.

This speculation is further studied by different characterizations. It has been revealed in Fe-P₇₀₀-PCC that P atoms could not be inserted into the graphitic carbon framework at 700 °C, all the P species are C-O-P and C-PO₂/C₂-PO₂ in the carbon surface, these P species cannot form stable coordination or interaction structure with Fe species, then the vast majority of Fe were washed off by hydrofluoric acid at template removing step, as expected, Fe-P₇₀₀-PCC was also inactive for hydrogenation reaction. When raising the carbonization temperature to above 800 °C, XPS and solid-state ³¹P NMR spectra indicate that part of P atoms has been inserted into the carbon framework to form graphitic phosphorus (0.21 atomic %). Furthermore, the EXAFS results also confirm the formation of Fe-P coordination. Gratifyingly, Fe-P₈₀₀-PCC gave a 69% conversion of quinoline. Elevating carbonization temperature to 900 °C produced the highest content of graphitic phosphorus (0.41 atomic %) and atomically dispersed single Fe atoms in the resultant Fe-P₉₀₀-PCC catalyst, which thus gave the best quinoline conversion of 93%. The contents of graphitic phosphorus declined to 0.35 and 0.21 atomic % when carbonization temperature is 1000 and 1100 °C, respectively, the conversion of quinoline accordingly decreased to 91% and 73%. Their EXAFS results indicate that some of single Fe atoms have transformed to Fe nanoparticles.

Based on above experimental investigations, it can be deduced that the graphitic P coordinated Fe is the active sites. To understand the atomic structure of catalytic site and the hydrogenation mechanism, the coordination structure of graphitic P coordinated single Fe atom has been investigated through quantitative EXAFS curve fitting and theoretical calculations. Besides, the H₂ absorption/activation on single atom Fe-P-C catalysts and the hydrogenation process of quinoline were studied by density

functional theory (DFT) calculations. These detailed explanations are presented below in the last response to the reviewer's comment.

References

1. Su, D. *et al. Angew. Chem. Int. Ed.* **57**, 13800-13804 (2018).
2. Zhang, J. *et al. ACS Appl. Mater. Interfaces*, **12**, 654-666 (2020).
3. Zhou, J-J. *et al. ChamCatChem.* **9**, 4287-4294 (2017).

Supplementary Fig. 15 Gas-phase isotopic H₂-D₂ exchange reactions over (a) P₉₀₀-PCC-polymer and Fe-P₉₀₀-PCC-polymer, and (b) Fe-P₉₀₀-PCC.

Comment 7. Moreover, the fitted P XPS spectra showed that no Fe-P state while XAFS results presented Fe-P state, it is contradictory.

Response 7. Thank you for this comment. In the P 2p XPS spectrum, the binding energy of Fe-P bond is in the range of 129-130 eV¹⁻². In this work, the relatively low content of Fe in the catalyst results in less Fe-P coordination on the surface of catalyst, and the content of Fe-P is lower than the detection limit of XPS, so it is not observed in the XPS analysis (as shown in Figure R3). On the other hand, the Fe k-edge EXAFS is for characterizing the coordination atoms of Fe and the detection limit of XAFS is much lower than XPS, so that Fe-P coordination could be detected by XAFS.

References

1. Zhang, J. *et al. Dalton Trans.* **46**, 16885-16894 (2017).
2. Wan, L. *et al. Inorg. Chem. Front.* **5**, 1094-1099 (2018).

Figure R3 The P 2p XPS spectrum of Fe-P₉₀₀-PCC.

Comment 8. *And the content of P in various samples should be measured by ICP, because of the incorrect content measured by XPS. It is complex that the content of P presented atomic % and the content of Fe present wt. %.*

Response 8. Thanks for this constructive suggestion. We have intended to measure the P content in Fe-P_x-PCC by ICP-MS. However, these carbon materials cannot be dissolved under many conditions, such as refluxing with concentrated HNO₃ or aqua regia. We agree with the reviewer that the XPS quantitative analysis is not very accurate, currently it is still a widely used method for quantitative measurement of light element in insoluble materials¹⁻². For accuracy, we regret to still use atomic % to represent content of P and use wt% to represent content of Fe.

References

1. Yu, J-S. *et al. J. Am. Chem. Soc.* **137**, 3165-3168 (2015).
2. Scherf, U. *et al. J. Am. Chem. Soc.* **142**, 2404-2412 (2020).

Comment 9. *If the chemical state of Fe-P₉₀₀-PCC is significant for the hydrogenation, the chemical state of the spent catalyst after the stability test is necessary.*

Response 9. Many thanks for your helpful suggestion. The spent Fe-P₉₀₀-PCC after eight repetitive uses has been characterized by ICP-MS, XPS, AC-STEM and XAS. ICP-MS analysis reveals that the Fe content is basically consistent with the fresh catalyst (0.069 wt% vs 0.071 wt%). Furthermore, the P 2p and Fe 2p XPS testify that the chemical states of graphitic P and Fe species keep constant (Supplementary Fig. 18a and 18b), and the content of graphitic P in Fe-P₉₀₀-PCC remains unchanged before and after hydrogenations (0.41 vs 0.41 atomic %, Supplementary Table 6). The AC-STEM image and EXAFS spectrum demonstrated that the single Fe atoms are preserved well (Supplementary Fig. 18c and 18d). All of these results demonstrate that the single Fe sites over Fe-P₉₀₀-PCC are stable under reaction conditions.

Accordingly, we have added below sentence into page 10 of the revised manuscript:

“P 2p and Fe 2p XPS spectra reveal that the chemical state of P_{grap} and Fe species keep constant (Supplementary Fig. 18 a and b), the content of P_{grap} remains unchanged at 0.41 atomic % before and after hydrogenations (Supplementary Table 6). The AC-STEM image and Fe k-edge EXAFS spectrum of spent Fe-P₉₀₀-PCC indicate that the atomically dispersed Fe species are well preserved after eight repetitive runs (Supplementary Fig. 18c and d).”

Supplementary Fig. 18 has been added into Supplementary Information:

Supplementary Fig. 18 Characterizations of spent Fe-P₉₀₀-PCC. **(a)** P 2p XPS spectra, the contents of P species are listed in Supplementary Table 6. **(b)** Fe 2p XPS spectra. **(c)** STEM image of Fe-P₉₀₀-PCC-used. **(d)** Fe k-edge EXAFS spectra of Fe-P₉₀₀-PCC and Fe-P₉₀₀-PCC-used, as well as the reference sample Fe foil.

Comment 10. *A comprehensive comparison with relative reports should be in Tables to demonstrate the catalytic performance.*

Response 10. Following the reviewer’s suggestion, we have compared the catalytic performance of Fe-P₉₀₀-PCC with previous reported non-noble metal catalysts in the hydrogenations of quinoline and nitrobenzene, as well as reductive amination reaction. As shown in Supplementary Table 11-13, the catalytic activity of current Fe-P₉₀₀-PCC catalyst is comparable to or much higher than a majority of previous reported heterogeneous catalysts in these reactions.

Accordingly, manuscript have been revised as follow (Page 9):

“The excellent catalytic performance of Fe-P₉₀₀-PCC is comparable or outperforms a majority of previous reported hydrogenation catalysts (as shown in Supplementary Table 11-13).”

Supplementary Table 11-13 have been added into Supplementary Information:

Supplementary Table 11. Catalytic performances for non-precious metal catalyzed heterogeneous hydrogenation of quinoline in earlier literatures.

Entry	Catalyst	Nanoparticle (NP) / Single atom (SA)	Reaction conditions	Yield (%)	TOF ^a (h ⁻¹)	Ref.
1	Fe-P ₉₀₀ -PCC	SA	150 °C, heptane, 4 MPa H ₂ , 12 h	92	60.4	This work
2	Co ₃ O ₄ -Co/NGr@α-Al ₂ O ₃	NP	120 °C, toluene, 2 MPa H ₂ , 48 h	98	0.5	1
3	Co ₁ /h-NC	SA	120 °C, THF, 3.5 MPa H ₂ , 10 h	56	5.6	2
4	Co@NGS-800-NL	NP	140 °C, isopropanol, 4 MPa H ₂ , 24 h	96	0.4	3
5	CoO _x @CN	NP	120 °C, methanol, 3.5 MPa H ₂ , 3 h	91	6.6	4
6	Fe(1)/L4(4.5)@C-800(12)	NP	130 °C, isopropanol-H ₂ O (3:1), 4 MPa H ₂ , 56 h	87	0.1	5
7	Ni NPs/[BMIM][Pro]	NP	75 °C, ethanol, 3 MPa H ₂ , 10 h	99	28.8	6

$${}^a\text{TOF} = \text{mol}_{\text{yield of tetrahydroquinoline}} / (\text{mol}_{\text{metal}} \cdot \text{h})$$

References

1. Chen, F. et al. *J. Am. Chem. Soc.* **137**, 11718-11724 (2015).
2. Huang, R. et al. *ACS Appl. Mater. Interfaces.* **12**, 17651-17658 (2020).
3. Li, F. et al. *J. Catal.* **355**, 53-62 (2017).
4. Wei, Z. et al. *ACS Catal.* **6**, 5816-5822 (2016).
5. Sahoo, B. et al. *Chem. Sci.* **9**, 8134-8141 (2018).
6. Sun, B. et al. *Catal. Lett.* **148**, 1336-1344 (2018).

Supplementary Table 12. Catalytic performances for non-precious metal catalyzed heterogeneous hydrogenation of nitrobenzene in earlier literatures.

Entry	Catalyst	Nanoparticle (NP) / Single atom (SA)	Reaction conditions	Yield (%)	TOF ^a (h ⁻¹)	Ref.
1	Fe-P ₉₀₀ -PCC	SA	100 °C, toluene, 4 MPa H ₂ , 18 h	99	43.7	This work
2	Fe-phen/C-800	NP	120 °C, H ₂ O-THF (1:1), 5 MPa H ₂ , 15 h	98	1.5	1
3	Co-L1/carbon	NP	110 °C, H ₂ O, 5 MPa H ₂ , 4 h	99	24.8	2
4	Co@mesoNC	SA	110 °C, ethanol, 3 MPa H ₂ , 2 h	55	42	3
5	Co-SiCN	NP	110 °C, ethanol-H ₂ O (4:1), 5 MPa H ₂ , 15 h	99	1.4	4
6	CoO _x @NCNTs	NP	110 °C, ethanol, 3 MPa H ₂ , 3 h	99	8.3	5
7	Co ₃ O ₄ /NGr@C	NP	110 °C, THF-H ₂ O (20:1), 5 MPa H ₂ , 4 h	95	25	6
8	Fe-N-C@CNTs-1.5	NP	110 °C, THF-H ₂ O (1:1), 5 MPa H ₂ , 6 h	99	46.8	7
9	Fe ₃ C@G-CNT-700	NP	40 °C, ethanol, 2 MPa H ₂ , 4.5 h	98	22	8
10	Fe/N-C-500	NP	120 °C, ethyl acetate, 4 MPa H ₂ , 15 h	99	0.6	9
11	Co-Co ₃ O ₄ @carbon-700	NP	110 °C, ethanol-H ₂ O (1:1), 4 MPa H ₂ , 15 h	99	3.9	10
12	Fe ₂ O ₃ @G-C-900	NP	70 °C, ethanol, 2 MPa H ₂ , 2 h	95	46.6	11
13	Co@NC-800	NP	110 °C, ethanol, 3 MPa H ₂ , 3 h	99	8	12
14	Co@NMC-800	NP	80 °C, ethanol, 1 MPa H ₂ , 80 min	99	37.5	13
15	Co ₂ P/CN _x	NP	60 °C, THF-H ₂ O (1:1), 5 MPa H ₂ , 6 h	99	1.5	14
16	Zr ₁₂ -TPDC-Co	SA	110 °C, toluene, 4 MPa H ₂ , 42 h	99	4.8	15
17	Ni/SiO ₂	NP	110 °C, ethanol, 2.5 MPa H ₂ , 7 h	99	1.2	16

18 ^a	Ni@PS ₆₀ SiCN	NP	110 °C, ethanol-H ₂ O (4:1), 5 MPa H ₂ , 20 h	99	5	17
19	7.2%Ni/Mo ₂ C	NP	80 °C, ethanol-H ₂ O (1:1), 2 MPa H ₂ , 1.5 h	99	32.3	18
20	Ni/C-300	NP	140 °C, ethanol, 2 MPa H ₂ , 2 h	71	17.7	19
21	Ni/AC _{OX}	NP	40 °C, toluene, 0.3 MPa H ₂ , 190 min	95	1.8	20
22	30.0 wt% Ni/C ₆₀ -Ac-B-4	NP	110 °C, ethanol, 2 MPa H ₂ , 5 h	99	6.3	21
23	Ni-NiO/NGr@C	NP	110 °C, THF-H ₂ O, 5 MPa H ₂ , 8 h	98	2.5	22
24	Ni/NGr@OMC-800	NP	100 °C, H ₂ O, 5 MPa H ₂ , 2 h	99	17.2	23
25	Ni-phen@SiO ₂ -1000	NP	40 °C, methanol-H ₂ O (1:1), 1 MPa H ₂ , 20 h	99	1.3	24

$$^a\text{TOF} = \text{mol}_{\text{yield of aniline}} / (\text{mol}_{\text{metal}} \cdot \text{h})$$

References

- Jagadeesh, R. V. et al. *Science*. **342**, 1073-1076 (2013).
- Weterhaus, F. A. et al. *Nature Chem.* **5**, 537-543 (2013).
- Gascon, J. et al. *J. Catal.* **357**, 20-28 (2018).
- Kempe, R. A. et al. *Angew. Chem. In. Ed.* **55**, 15175-15179 (2016).
- Wei, Z. et al. *ACS Catal.* **5**, 4783-4789 (2015).
- Jagadeesh R. V. et al. *Nature Protocols.* **10**, 916-926 (2015).
- Chen, J. et al. *RSC Adv.* **6**, 96203-96209 (2016).
- Hou, Z. et al. *Carbon*, **99**, 330-337 (2016).
- Sheng, H. et al. *RSC Adv.* **6**, 96431-96435 (2016).
- Sahoo, B. et al. *ChemSusChem.* **10**, 3035-3039 (2017).
- Hou, Z. et al. *Chin. J. Catal.* **38**, 1909-1917 (2017).
- Sun, X et al. *ChemCatChem.* **9**, 1854-1862 (2017).
- Zhang, X-M. et al. *J. Catal.* **348**, 212-222 (2017).
- Yang, S. et al. *Chem. -A. Euro. J.* **24**, 4234-4238 (2018).
- Ji, P. et al. *J. Am. Chem. Soc.* **139**, 7004-7011 (2017).
- Zheng, Y. et al. *Catal. Lett.* **124**, 268-276 (2008).
- Kempe, R. et al. *ChemCatChem.* **8**, 2461-2465 (2016).
- Shu, Y. et al. *Appl. Surf. Sci.* **396**, 339-346 (2017).

19. Zhang, P. et al. *Phys. Chem. Chem. Phys.* **17**, 145-150 (2015).
 20. Zhang, T. et al. *Chem. Comm.* **53**, 1969-1972 (2017).
 21. Wang, G. et al. *Catal. Comm.* **97**, 83-87 (2017).
 22. Pisiewicz, S. et al. *ChemCatChem*. **8**, 129-134 (2016).
 23. Huang, H. et al. *RSC Adv.* **8**, 8898-8909 (2018).
 24. Ryabchuk, P et al. *Sci. Adv.* **4**, eaat0761 (2018).

Supplementary Table 13. Catalytic performances for non-precious metal catalyzed heterogeneous reductive amination of carbonyl compounds in earlier literatures.

Entry	Catalyst	Nanoparticle (NP) / Single atom (SA)	Reaction conditions	Yield (%)	TOF ^a (h ⁻¹)	Ref.
1	Fe-P ₉₀₀ -PCC	SA	75 °C, H ₂ O, 6 MPa H ₂ , 30 h	98	173	This work
	Reductive amination of aldehyde for the synthesis of primary amine: 					
2	Fe-P ₉₀₀ -PCC	SA	140 °C, H ₂ O, 6 MPa H ₂ , 20 h	98	193	This work
						
3	Co-DABCO-TPA@C-800	NP	120 °C, t-BuOH, 4 MPa H ₂ , 15 h	88	1.7	1
	Reductive amination of aldehyde for the synthesis of primary amine: 					
4	Co-DABCO-TPA@C-800	NP	120 °C, t-BuOH, 4 MPa H ₂ , 24 h	87	1.0	1
	Reductive amination of aldehyde for the synthesis of tertiary amine: 					
5	Ni-TA@SiO ₂ -800	NP	120 °C, t-BuOH, 2 MPa H ₂ , 24 h	98	0.7	2
	Reductive amination of aldehyde for the synthesis of primary amine: 					
6	Ni/gama-Al ₂ O ₃	NP	80 °C, H ₂ O, 1 MPa H ₂ , 20 h	99	4.2	3
	Reductive amination of aldehyde for the synthesis of primary amine: 					
7	Fe/(N)SiC	NP	130 °C, H ₂ O, 6.5 MPa H ₂ , 20 h	89	0.4	4

	Reductive amination of aldehyde for the synthesis of primary amine:					
						
8	Fe/(N)SiC	NP	140 °C, H ₂ O, 6.5 MPa H ₂ , 20 h	99	0.5	
	Reductive amination of ketone for the synthesis of primary amine:					
						
9	Co/N-C-800	NP	110 °C, H ₂ O, 0.5 MPa H ₂ , 4 h	92	1.8	5
	Reductive amination of aldehyde for the synthesis of primary amine:					
						
10	Raney Ni	-	120 °C, methanol, 1 MPa H ₂ , 2 h	65	1.0	6
	Raney Co	-	120 °C, methanol, 1 MPa H ₂ , 2 h	98	3.1	
	Reductive amination of furfural for the synthesis of primary amine:					
						
11	Ni ₆ AlO _x	NP	100 °C, H ₂ O, 0.1 MPa H ₂ , 6 h	99	0.3	7
						
12	Co@NC-800	NP	130 °C, ethanol, 1 MPa H ₂ , 12 h	97	11.9	8

$${}^a\text{TOF} = \text{mol}_{\text{yield of product}} / (\text{mol}_{\text{metal}} \cdot \text{h})$$

References

- Jagadeesh, R. V. et al. *Science*. **358**, 326-332 (2017).
- Jagadeesh, R. V. et al. *Angew. Chem. Int. Ed.* **58**, 5064-5068 (2019).
- Kempe, R. et al. *Nature Catal.* **2**, 71-77 (2019).
- Bauer, C. et al. *ChemSusChem*. **10**.1002/cssc.20200856 (2020).
- Zhang, Z. et al. *Arab. J. Chem.* **13**, 4916-4925 (2020)
- Wei, J. et al. *ChemCatChem*. **11**, 5562-5569 (2019).
- Yuan, H. et al. *ACS Omega*. **4**, 2510-2516 (2019).
- Chi, Q. et al. *J. Catal.* **370**, 347-356 (2019).

Comment 11. *Conclusions should not be involved in discussion and theoretical calculation of active sites is suggested here.*

Response 11. Thank you very much for this comment. Following your and Editor's suggestion, we have merged the "Discussion" to the "Results" to form a single section entitled "Results and Discussion".

Furthermore, the coordination structure of graphitic P coordinated single Fe atom has been investigated through quantitative EXAFS curve fitting and theoretical calculations. Besides, the reaction pathway of hydrogenation of quinoline over Fe-P₉₀₀-PCC catalyst have been studied by density functional theory (DFT) calculations.

For the representative Fe-P₉₀₀-PCC catalyst, its EXAFS spectrum shows that the main peak locates at 1.63 Å, ascribing to Fe-P first shell coordination. Furthermore, the Fe-O first shell coordination at 1.45 Å is also included in this broadening peak, which indicates that O need to be included in the curve fitting. On the other hand, a shoulder peak at 2.55 Å for Fe-C second shell coordination is also observed. Therefore, a three-shell structure model, including a Fe-P, a Fe-O and a Fe-C shell, was initially used to fit the EXAFS data of Fe-P₉₀₀-PCC. The best-fitting analyses manifests that the dominant contribution is given by Fe-P and Fe-O first shell coordination as presented in Fig. 3c and 3d. The coordination numbers for P and O atoms are calculated as 4.0 ± 0.8 and 2.0 ± 0.4 , and the corresponding mean bond length of Fe-P and Fe-O are 2.35 ± 0.02 Å and 2.00 ± 0.03 Å, respectively. These results reveal that the single Fe atoms in Fe-P₉₀₀-PCC coordinate with four P atoms and a dioxygen molecule (O₂-Fe-P₄). Because the atomic size of P (106 pm) is larger than C (75 pm), Fe center adopts a pyramidal geometry as shown in Fig. 3e, this structure is quite different from the planar structure of Fe-N₄. Besides, the theoretical calculations reveal that the binding energy of O₂-Fe-P₄ structure is -2.09 eV, suggesting this coordination structure is thermodynamically stable.

To further understand the hydrogenation reaction process at the atomic level, DFT calculations were performed to study the reaction mechanism (hydrogenation of quinoline was choose as model reaction). The DFT calculations reveal that the original O₂-Fe-P₄ sites on Fe-P₉₀₀-PCC can be reduced to the Fe-P₄ sites in the presence of H₂ (the reaction of O₂-Fe-P₄ + 2H₂ → Fe-P₄ + 2H₂O is exothermic by 4.08 eV), the H₂ is then adsorbed on Fe atom and the H-H bond length is elongated to 0.81 Å with a free energy of -0.407 eV, suggesting that the H₂ can be catalytically activated on the Fe-P₄ active sites originated from present M-P-C catalyst.

Accordingly, we have revised our manuscript in Page 6 as the following:

“The coordination information of Fe atom in optimal Fe-P₉₀₀-PCC is further studied using quantitative EXAFS curve fitting (see Supplementary Table 7 for details). The best-fitting analyses is shown in Fig. 3c and 3d, which manifest that the Fe atom is coordinated by four P atoms and a

dioxygen molecule (O_2 -Fe- P_4), forming a pyramidal geometry as shown in Fig. 3e, this structure is quite different from the planar structure of Fe- N_4 . Besides, the theoretical calculations reveal that the binding energy of O_2 -Fe- P_4 structure is -2.09 eV, suggesting this coordination structure is thermodynamically stable.”

The revised Fig. 3 has been added into page 6 of the revised manuscript as below:

Fig. 3 Characterizations of Fe species. **a** EXAFS spectra of Fe- P_x -PCCs, $Fe_{0.4}/Fe-P_{900}$ -PCC, and reference materials (Fe foil and Fe_2O_3). **b** ^{57}Fe Mössbauer spectrum of Fe- P_{900} -PCC. **c** and **d** The experimental Fe k -edge EXAFS spectrum (blue dot) and the fitting curve (red line) of Fe- P_{900} -PCC at k and R space, respectively. **e** Top and side views of optimized structural model of O_2 -Fe- P_4 .

Supplementary Table 7 has been added in Supplementary Information:

Supplementary Table 7. Fitting results of Fe k -edge EXAFS data for Fe@Fe- N_{900} -PCC^a and Fe- P_{900} -PCC^b.

Sample	Bond	CN	R (Å)	σ^2 (Å ²)	ΔE_0 (eV)	R-factor
Fe@Fe- N_{900} -PCC	Fe-N	4.2 ± 0.4	1.95 ± 0.01	0.007 ± 0.001	-6.23 ± 1.1	0.004
Fe- P_{900} -PCC	Fe-P	4.0 ± 0.8	2.35 ± 0.02	0.014 ± 0.006	-0.96 ± 0.3	0.013
	Fe-O	2.0 ± 0.4	2.00 ± 0.03	0.004 ± 0.004	-0.96 ± 0.3	

The average lengths of Fe-N, Fe-P and Fe-O bonds and coordination numbers of Fe atoms are extracted from the curve fitting for Fe k -edge EXAFS data (Supplementary Fig. 13 and Manuscript Fig. 3). CN, coordination number; R, distance between absorber and backscatter atoms; σ^2 , the Debye-Waller factor; ΔE_0 , inner potential correction; R-factor, indicate the goodness of the fit.

^aFor the EXAFS spectrum of Fe@Fe- N_{900} -PCC (Supplementary Fig. 12), only a strong Fe-N peak at

1.51 Å is observed. So, the fitting was performed by including a single Fe-N shell within the R-rang of 1 - 3.1 Å and *k*-rang of 1.42 Å⁻¹ - 9.62 Å⁻¹. The fitting results reveal that the coordination number of Fe center with surrounding N atoms is 4.2 ± 0.4 and the average Fe-N bond length is 1.95 Å ± 0.01, suggesting the single Fe sites in Fe@Fe-N₉₀₀-PCC adopt a planar Fe-N₄ structure (as presented in Supplementary Fig. 13b).

^bThe EXAFS spectrum of Fe-P₉₀₀-PCC shows that the main peak locates at 1.63 Å, ascribing to Fe-P first shell coordination. Furthermore, the Fe-O first shell coordination at 1.45 Å is also included in this broadening peak, which indicates that O need to be included in the curve fitting. On the other hand, a shoulder peak at 2.55 Å for Fe-C second shell coordination is also observed. Therefore, a three-shell structure model, including a Fe-P, a Fe-O and a Fe-C shell, is initially used to fit the EXAFS data of Fe-P₉₀₀-PCC within the R-rang of 1 - 3.1 Å and *k*-rang of 1.42 Å⁻¹ - 9.62 Å⁻¹. The best-fitting analyses manifests that the dominant contribution is given by Fe-P and Fe-O first shell coordination as presented in Manuscript Fig. 3c and 3d. The coordination numbers for P and O atoms are calculated as 4.0 ± 0.8 and 2.0 ± 0.4, and the corresponding mean bond length of Fe-P and Fe-O are 2.35 ± 0.02 and 2.00 ± 0.03 Å, respectively. These results reveal that the single Fe atom in Fe-P₉₀₀-PCC coordinates with four P atoms and a dioxygen molecule (O₂-Fe-P₄). Because the atomic size of P (106 pm) is larger than C (75 pm), Fe center adopts a pyramidal geometry as shown in Manuscript Fig. 3e, this structure is quite different from the planar structure of Fe-N₄.

We have revised our manuscript in Page 7 as below:

“To understand the possible mechanism for the hydrogenation of quinoline, we studied the reaction process with DFT calculations, which initially reveals that the bond length of O-O in O₂-Fe-P₄ is 1.314 Å, the two oxygen atoms have a total magnetic moment of 0.77 uB, O₂ gains approximately one electron from Fe-P₄ upon adsorption, and the adsorbed O₂ is assigned as O₂⁻ which can be readily reduced by H₂ since the reaction (O₂-Fe-P₄ + 2H₂ → Fe-P₄ + 2H₂O) is exothermic by 4.08 eV. Therefore, the Fe-P₄ was used as the starting point to calculate the reaction pathway of hydrogenation of quinoline. As shown in Fig. 4, the H₂ preferentially adsorbs on Fe atom, the bond length of H-H is elongated to 0.81 Å with a free energy of -0.407 eV, indicating that the H₂ can easily adsorb and bond on Fe-P₄. On the other hand, the adsorption energy of quinoline on Fe atoms is -0.687 eV (Supplementary Table 9 and 10). Then, one hydrogen is transferred to N atom (C₉H₈N*), leaving another hydrogen bound to Fe atom (int-2 to int-3). Hydrogenation of C₉H₈N* to adsorbed C₉H₉N* proceeds with an energy barrier of 0.38 eV and is exothermic by -0.348 eV (from int-3 to int-4). The reaction continues to give final product via the addition of second hydrogen molecule, and the reaction of C₉H₉N* to C₉H₁₀N* (int-5 to int-6) is found to be the rate determining step with an energy barrier of 0.728 eV.”

The above mentioned Fig. 4 is added into the revised manuscript as below:

Fig. 4 Energy profile of hydrogenation of quinoline over Fe-P₉₀₀-PCC.

Supplementary Table 9 and 10 have been added in Supplementary Information:

Supplementary Table 9. Step by step barrier (E_a , eV) and reaction energy (E_r , eV) for hydrogenation of quinoline (C_9H_7N) over Fe-P₉₀₀-PCC.

Number	Reactions	E_a (eV)	E_r (eV)
1	$H_2(g) \rightarrow H_2^*$	-	-0.407
2	$C_9H_7N + H_2^* \rightarrow C_9H_7N^* + H_2^*$	-	-0.687
3	$C_9H_7N^* + H_2^* \rightarrow C_9H_8N^* + H(Fe)^*$	0.220	-0.004
4	$C_9H_8N^* + H(Fe)^* \rightarrow C_9H_9N^*$	0.380	-0.348
5	$C_9H_9N^* + H_2(g) \rightarrow C_9H_9N^* + H_2^*$	-	-0.025
6	$C_9H_9N^* + H_2^* \rightarrow C_9H_{10}N^* + H(Fe)^*$	0.728	0.432
7	$C_9H_{10}N^* + H(Fe)^* \rightarrow C_9H_{11}N$	0.132	-1.331

Supplementary Table 10. The energies of species in the processes of hydrogenation of quinoline (C_9H_7N).

Label	Species	E (eV)	E_{rel} (eV) ^a
IS	$C_9H_7N + H_2(g)$	-635.885	0.000
int-1	$C_9H_7N + H_2^*$	-643.051	-0.407
int-2	$C_9H_7N^* + H_2^*$	-758.800	-1.094
TS1	-	-	-
int-3	$C_9H_8N^* + H(Fe)^*$	-758.804	-1.098
TS2	-	-	-
int-4	$C_9H_9N^*$	-759.152	-1.446
int-5	$C_9H_9N^* + H_2^*$	-765.936	-1.471
TS3	-	-	-
int-6	$C_9H_{10}N^* + H(Fe)^*$	-765.504	-1.039
TS4	-	-	-
int-7	$C_9H_{11}N$	-766.836	-2.370
FS	-	-635.885	-1.801

^aThe E_{rel} refers to the energy of species labelled IS.

Reviewer #3

Comment. *In this manuscript, the authors reported that graphitic phosphorus species (P_{grap}) coordinated single-atom Fe on are active in hydrogenation of N-heterocycles, functionalized nitroarenes, and reductive amination reactions, while single-atom Fe atoms on either carbon or N-doped carbon are almost inactive. XPS was employed to verify the correlations between the graphic P and hydrogenation activity. XAFS and STEM confirmed the atomic presence of Fe. The manuscript might be publishable in this high standard journal after addressing the following questions:*

Response. We appreciate the reviewer very much for the positive comment.

Comment 1. *One of the major drawbacks of this manuscript is lack of the atomic structure of this Fe-P-C catalyst. What are the coordination numbers of Fe-P? How Fe is coordinated with graphic P? What is the structure?*

Response 1. We appreciate this insightful comment very much. Following your and other reviewer's advice, the atomic structure of active sites in Fe-P₉₀₀-PCC has been investigated through quantitative EXAFS curve fitting and theoretical calculations.

Based on the EXAFS spectrum of Fe-P₉₀₀-PCC, a three-shell structure model including a Fe-O (1.45 Å, first shell coordination), a Fe-P (1.63 Å, first shell coordination) and a Fe-C (2.55 Å, second shell coordination) shell was used to fit the EXAFS data of this sample. The best-fitting results indicate that the dominant contribution is given by Fe-P and Fe-O first shell coordination (Fig. 3c and 3d). The coordination numbers for P and O atoms were calculated as 4.0 ± 0.8 and 2.0 ± 0.4 , and the corresponding mean bond length of Fe-P and Fe-O are 2.35 ± 0.02 and 2.00 ± 0.03 Å, respectively (Supplementary Table 7). These results revealed that the single Fe sites in Fe-P₉₀₀-PCC are coordinated with four P atoms and a dioxygen molecule ($O_2\text{-Fe-P}_4$). Because the atomic size of P (106 pm) is larger than C (75 pm), Fe center adopts a pyramidal geometry as shown in Fig. 3e. Besides, the theoretical calculations reveal that the binding energy of $O_2\text{-Fe-P}_4$ structure is -2.09 eV, suggesting this coordination structure is thermodynamically stable.

Subsequently, we have revised our manuscript in Page 6 as the following:

“The coordination information of Fe atom in optimal Fe-P₉₀₀-PCC is further studied using quantitative EXAFS curve fitting (see Supplementary Table 7 for details). The best-fitting analyses is shown in Fig. 3c and 3d, which manifest that the Fe atom is coordinated by four P atoms and a dioxygen molecule ($O_2\text{-Fe-P}_4$), forming a pyramidal geometry as shown in Fig. 3e, this structure is quite different from the planar structure of Fe-N₄. Besides, the theoretical calculations reveal that the

binding energy of O₂-Fe-P₄ structure is -2.09 eV, suggesting this coordination structure is thermodynamically stable.”

The revised Fig. 3 has been added into page 6 of the revised manuscript as below:

Fig. 3 Characterizations of Fe species. **a** EXAFS spectra of Fe-P_x-PCCs, Fe_{0.4}/Fe-P₉₀₀-PCC, and reference materials (Fe foil and Fe₂O₃). **b** ⁵⁷Fe Mössbauer spectrum of Fe-P₉₀₀-PCC. **c** and **d** The experimental Fe k-edge EXAFS spectrum (blue dot) and the fitting curve (red line) of Fe-P₉₀₀-PCC at *k* and *R* space, respectively. **e** Top and side views of optimized structural model of O₂-Fe-P₄.

Supplementary Table 7 has been added in Supplementary Information:

Supplementary Table 7. Fitting results of Fe k-edge EXAFS data for Fe@Fe-N₉₀₀-PCC^a and Fe-P₉₀₀-PCC^b.

Sample	Bond	CN	R (Å)	σ^2 (Å ²)	ΔE_0 (eV)	R-factor
Fe@Fe-N ₉₀₀ -PCC	Fe-N	4.2 ± 0.4	1.95 ± 0.01	0.007 ± 0.001	-6.23 ± 1.1	0.004
	Fe-P	4.0 ± 0.8	2.35 ± 0.02	0.014 ± 0.006	-0.96 ± 0.3	0.013
Fe-P ₉₀₀ -PCC	Fe-O	2.0 ± 0.4	2.00 ± 0.03	0.004 ± 0.004	-0.96 ± 0.3	

The average lengths of Fe-N, Fe-P and Fe-O bonds and coordination numbers of Fe atoms are extracted from the curve fitting for Fe k-edge EXAFS data (Supplementary Fig. 13 and Manuscript Fig. 3). CN, coordination number; R, distance between absorber and backscatter atoms; σ^2 , the Debye-Waller factor; ΔE_0 , inner potential correction; R-factor, indicate the goodness of the fit.

^aFor the EXAFS spectrum of Fe@Fe-N₉₀₀-PCC (Supplementary Fig. 12), only a strong Fe-N peak at 1.51 Å is observed. So, the fitting was performed by including a single Fe-N shell within the R-rang of 1 - 3.1 Å and *k*-rang of 1.42 Å⁻¹ - 9.62 Å⁻¹. The fitting results reveal that the coordination number of

Fe center with surrounding N atoms is 4.2 ± 0.4 and the average Fe-N bond length is $1.95 \text{ \AA} \pm 0.01$, suggesting the single Fe sites in Fe@Fe-N₉₀₀-PCC adopt a planar Fe-N₄ structure (as presented in Supplementary Fig. 13b).

^bThe EXAFS spectrum of Fe-P₉₀₀-PCC shows that the main peak locates at 1.63 Å, ascribing to Fe-P first shell coordination. Furthermore, the Fe-O first shell coordination at 1.45 Å is also included in this broadening peak, which indicates that O need to be included in the curve fitting. On the other hand, a shoulder peak at 2.55 Å for Fe-C second shell coordination is also observed. Therefore, a three-shell structure model, including a Fe-P, a Fe-O and a Fe-C shell, is initially used to fit the EXAFS data of Fe-P₉₀₀-PCC within the R-rang of 1 - 3.1 Å and k-rang of 1.42 Å⁻¹ - 9.62 Å⁻¹. The best-fitting analyses manifests that the dominant contribution is given by Fe-P and Fe-O first shell coordination as presented in Manuscript Fig. 3c and 3d. The coordination numbers for P and O atoms are calculated as 4.0 ± 0.8 and 2.0 ± 0.4 , and the corresponding mean bond length of Fe-P and Fe-O are 2.35 ± 0.02 and 2.00 ± 0.03 Å, respectively. These results reveal that the single Fe atom in Fe-P₉₀₀-PCC coordinates with four P atoms and a dioxygen molecule (O₂-Fe-P₄). Because the atomic size of P (106 pm) is larger than C (75 pm), Fe center adopts a pyramidal geometry as shown in Manuscript Fig. 3e, this structure is quite different from the planar structure of Fe-N₄.

Comment 2. *The authors might also add some additional discussion about the hydrogenation mechanism.*

Response 2. Thank you very much for the very important comment. Following this suggestion, we have performed DFT calculations to investigate the reaction mechanism of the hydrogenation of quinoline over Fe-P₉₀₀-PCC. A model of the O₂-Fe-P₄ structure embedded in graphene was adapted according to the experimental characterizations and theoretical calculations. Under the H₂ atmosphere, the DFT calculations demonstrated that the Fe atom coordinated dioxygen can be readily removed by H₂ (the reaction (O₂-Fe-P₄ + 2H₂ → Fe-P₄ + 2H₂O) is exothermic by 4.08 eV). After that, the in-situ generated Fe-P₄ structure can efficiently absorb and activate H₂ molecule, thereby catalyze the hydrogenation reaction. The calculated reaction pathway of hydrogenation of quinoline is shown in Fig. 4.

We have revised our manuscript in Page 7 as below:

“**DFT calculations.** To understand the possible mechanism for the hydrogenation of quinoline, we studied the reaction process with DFT calculations, which initially reveals that the bond length of O-O in O₂-Fe-P₄ is 1.314 Å, the two oxygen atoms have a total magnetic moment of 0.77 uB, O₂ gains approximately one electron from Fe-P₄ upon adsorption, and the adsorbed O₂ is assigned as O₂⁻ which can be readily reduced by H₂ since the reaction (O₂-Fe-P₄ + 2H₂ → Fe-P₄ + 2H₂O) is exothermic by

4.08 eV. Therefore, the Fe-P₄ was used as the starting point to calculate the reaction pathway of hydrogenation of quinoline. As shown in Fig. 4, the H₂ preferentially adsorbs on Fe atom, the bond length of H-H is elongated to 0.81 Å with a free energy of -0.407 eV, indicating that the H₂ can easily adsorb and bond on Fe-P₄. On the other hand, the adsorption energy of quinoline on Fe atoms is -0.687 eV (Supplementary Table 9 and 10). Then, one hydrogen is transferred to N atom (C₉H₈N*), leaving another hydrogen bound to Fe atom (int-2 to int-3). Hydrogenation of C₉H₈N* to adsorbed C₉H₉N* proceeds with an energy barrier of 0.38 eV and is exothermic by -0.348 eV (from int-3 to int-4). The reaction continues to give final product via the addition of second hydrogen molecule, and the reaction of C₉H₉N* to C₉H₁₀N* (int-5 to int-6) is found to be the rate determining step with an energy barrier of 0.728 eV.”

The above mentioned Fig. 4 is added into the revised manuscript as below:

Fig. 4 Energy profile of hydrogenation of quinoline over Fe-P₉₀₀-PCC.

Supplementary Table 9 and 10 have been added in Supplementary Information:

Supplementary Table 9. Step by step barrier (E_a , eV) and reaction energy (E_r , eV) for hydrogenation of quinoline (C_9H_7N) over Fe-P₉₀₀-PCC.

Number	Reactions	E_a (eV)	E_r (eV)
1	$H_2(g) \rightarrow H_2^*$	-	-0.407
2	$C_9H_7N + H_2^* \rightarrow C_9H_7N^* + H_2^*$	-	-0.687
3	$C_9H_7N^* + H_2^* \rightarrow C_9H_8N^* + H(Fe)^*$	0.220	-0.004
4	$C_9H_8N^* + H(Fe)^* \rightarrow C_9H_9N^*$	0.380	-0.348
5	$C_9H_9N^* + H_2(g) \rightarrow C_9H_9N^* + H_2^*$	-	-0.025
6	$C_9H_9N^* + H_2^* \rightarrow C_9H_{10}N^* + H(Fe)^*$	0.728	0.432
7	$C_9H_{10}N^* + H(Fe)^* \rightarrow C_9H_{11}N$	0.132	-1.331

Supplementary Table 10. The energies of species in the processes of hydrogenation of quinoline (C_9H_7N).

Label	Species	E (eV)	E_{rel} (eV) ^a
IS	$C_9H_7N + H_2(g)$	-635.885	0.000
int-1	$C_9H_7N + H_2^*$	-643.051	-0.407
int-2	$C_9H_7N^* + H_2^*$	-758.800	-1.094
TS1	-	-	-
int-3	$C_9H_8N^* + H(Fe)^*$	-758.804	-1.098
TS2	-	-	-
int-4	$C_9H_9N^*$	-759.152	-1.446
int-5	$C_9H_9N^* + H_2^*$	-765.936	-1.471
TS3	-	-	-
int-6	$C_9H_{10}N^* + H(Fe)^*$	-765.504	-1.039
TS4	-	-	-
int-7	$C_9H_{11}N$	-766.836	-2.370
FS	-	-635.885	-1.801

^aThe E_{rel} refers to the energy of species labelled IS.

Comment 3. *On page 6, the authors claimed that “the formation of P_{grap} is the prerequisite for the existence of single Fe atoms on the surface of P-doped porous carbon”. What is the evidence for this? Then what is the form of the Fe in Fe-P₇₀₀-PCC?*

Response 3. Thank you very much for this valuable comment. Your concern on the relationships between graphitic phosphorus and single Fe atoms is important, which helps us to give a clearer explanation for the formation mechanism of active sites.

At 700 °C, P atoms cannot insert into graphitic carbon framework, all of the P species are PO_x (determined by XPS and solid-state ^{31}P NMR), these P functional groups are unstable and cannot form stable structure with Fe atoms, the vast majority of Fe came from raw materials were removed by hydrofluoric acid at template removing step. When increasing the carbonization temperature to above 800 °C, the results of XPS and solid-state ^{31}P NMR indicate that a partial of P atoms have inserted into the carbon framework to form graphitic phosphorus. Furthermore, the EXAFS results confirm that Fe-P coordination have formed when temperature is above 800 °C. The higher content of graphitic phosphorus will coordinate more single Fe atoms on the surface of porous carbon material. For example, the content of graphitic phosphorus of Fe-P₉₀₀-PCC is the highest (0.41 atomic %), most of Fe species are atomically dispersed single Fe atoms. When further increasing carbonization temperature to 1000 and 1100 °C, the contents of graphitic phosphorus are decreased to 0.35 and 0.21 atomic %, respectively. The EXAFS results indicate that some of single Fe atoms have transformed to Fe nanoparticles. Based on above investigations, we deduced that the single Fe atoms are coordinated and stabilized by graphitic phosphorus.

For Fe-P₇₀₀-PCC, we have intended to investigate the chemical state of Fe species in this sample through X-ray absorption spectroscopy (XAS). Unfortunately, since the Fe content of Fe-P₇₀₀-PCC is very low (0.0072 wt%), the signal of Fe in X-ray absorption near edge structure (XANES) is very weak and cannot get useful information from this characterization (Figure R1). As discussed above, we have demonstrated that there is no graphitic P coordinated single Fe atom on the surface of Fe-P₇₀₀-PCC, so we deduced that the very low content of Fe presented in this sample is from the carbon layers encapsulated Fe aggregates.

Figure R1. The normalized XANES spectra (a) and Fe k-edge EXAFS spectra (b) of Fe-P₇₀₀-PCC and the reference materials.

Comment 4. *On page 5, the authors said that “even though the AC-STEM has observed the presence of some Fe nanoparticles in Fe-P₉₀₀-PCC (Supplementary Fig. 10), these nanoparticles are deeply embedded in the carbon matrix and could not be detected by the high-energy X-ray.” This is not correct, since XAFS is a bulk technique.*

Response 4. Thanks for your valuable suggestion which promotes us to improve the manuscript quality. We are sorry for this incorrect statement. Indeed, XAFS is a bulk technique and the embedded nanoparticle is detectable. In this work, the vast majority of Fe species are existed in the form of single atom, the proportion of carbon shell encapsulated Fe nanoparticles is extremely low (as shown in Figure R2). Therefore, it is reasonable that these small amounts of Fe nanoparticles are not detectable in XAFS, but they can be observed by AC-STEM.

Figure R2. TEM images of Fe-P₉₀₀-PCC. Fe nanoparticles are highlighted by yellow circles.

We have accordingly made corrections in Page 3 of revised manuscript as below:

“Additionally, some of carbon shells encapsulated Fe nanoparticles are also observed in these carbon materials (Supplementary Fig. 9 and 10), but no Fe-Fe coordination have been detected in characterization of Fe k-edge extended X-ray absorption fine structure (EXAFS, which will be discussed below), suggesting the majority of Fe species are existed in the form of single atom, and the proportion of carbon shells encapsulated Fe aggregates is extremely low.”

Comment 5. *AC-STEM images of Fe-C₉₀₀-PCC are missing. They be useful to confirm the presence of Fe atoms in this sample.*

Response 5. Thank you very much for this suggestive comment. In this work, we have synthesized Fe-C₉₀₀-PCC and Fe@Fe-C₉₀₀-PCC as reference catalysts to investigate non-heteroatom doped carbon supported single Fe atoms does have activity for hydrogenation reaction. To facilitate characterization and make the results more convincing, Fe@Fe-C₉₀₀-PCC was chosen as representative sample. The AC-STEM (Supplementary Fig. 11 a and b) characterization show that the single Fe atoms are dispersed on the surface of porous carbon, and Fe nanoparticles are also observed in the HRTEM image (Supplementary Fig. 11c). In addition, the Fe k-edge EXAFS analysis also suggests that both Fe single atoms and nanoparticles are dispersed on Fe@Fe-C₉₀₀-PCC, since the main peak at 1.45 Å assigning to Fe-C coordination and small peak of Fe-Fe coordination at 2.19 Å are all observed (Supplementary Fig. 11d).

Accordingly, we have revised the manuscript in Page 4 as below:

“AC-STEM and HRTEM images as well as Fe k-edge EXAFS spectrum reveal that both Fe single atoms and nanoparticles are dispersed on Fe@Fe-C₉₀₀-PCC (Supplementary Fig. 11).”

The above mentioned Supplementary Fig. 11 has been added into the Supplementary Information:

Supplementary Fig. 11 Characterizations of Fe@Fe-C₉₀₀-PCC: (a) and (b) AC-STEM images, Fe single atoms are highlighted by yellow circles, (c) HRTEM image, and (d) EXAFS spectra of Fe-N₉₀₀-PCC and reference materials (Fe foil and Fe₂O₃).

Comment 6. *On page 8, it should be “summary or conclusion” rather than “discussion”.*

Response 6. Thanks very much for this comment. Following your and Editor’s suggestion, we have merged the “Discussion” to the “Results” to form a single section entitled “Results and Discussion”.

Comment 7. *This reviewer is wondering what the activities of Fe nanoparticles on carbon or N-doped carbon supports are, it might be useful to make the comparison as well.*

Response 7. Thank you very much for this suggestive comment. Following your suggestion, we have synthesized Fe_{1.0}/Fe-C₉₀₀-PCC and Fe_{1.0}/Fe-N₉₀₀-PCC through impregnation method by using Fe-C₉₀₀-PCC and Fe-N₉₀₀-PCC as the support, respectively. The preparation process is as follow: 0.5 g Fe-C₉₀₀-PCC (or Fe-N₉₀₀-PCC) is dispersed in 5 mL of ultrapure water, and then the suspension is

stirred at room temperature for 15 minutes. Subsequently, aqueous $\text{Fe}(\text{NO}_3)_3$ solution (1 mL) containing 40 mg $\text{Fe}(\text{NO}_3)_3 \cdot 9\text{H}_2\text{O}$ is added dropwise to the suspension. After stirred at room temperature for 12 hours, the water of suspension was evaporated at $70\text{ }^\circ\text{C}$. The obtained powder was dried under vacuum at $60\text{ }^\circ\text{C}$ for 8 hours, and then calcined at $450\text{ }^\circ\text{C}$ for 2 hours under N_2 .

The TEM characterizations indicate that there are a number of Fe nanoparticles on the surface of $\text{Fe}_{1.0}/\text{Fe-C}_{900}\text{-PCC}$ and $\text{Fe}_{1.0}/\text{Fe-N}_{900}\text{-PCC}$ (Figure R3). Then, their catalytic performances in the hydrogenations of quinoline and nitrobenzene have been assessed. As listed in Table R1 and Table R2, both of them have very low catalytic activities in these two hydrogenation reactions, which are similar to that of $\text{Fe}_{0.95}/\text{Fe-P}_{900}\text{-PCC}$.

Figure R3. TEM images: (a) and (b) $\text{Fe}_{1.0}/\text{Fe-C}_{900}\text{-PCC}$, (c) and (d) $\text{Fe}_{1.0}/\text{Fe-N}_{900}\text{-PCC}$. Fe nanoparticles are highlighted by yellow arrows.

Table R1. Hydrogenation of quinoline catalyzed by Fe_{1.0}/Fe-C₉₀₀-PCC, Fe_{1.0}/Fe-N₉₀₀-PCC and Fe_{0.95}/Fe-P₉₀₀-PCC.

Reaction scheme showing the hydrogenation of quinoline (1) to 1,2,3,4-tetrahydroquinoline (2) using a catalyst and 4 MPa H₂.

Entry	Catalyst	Temperature (°C)	Conversion (%)	Yield (%)
1	Fe _{1.0} /Fe-C ₉₀₀ -PCC	150	5	5
2	Fe _{1.0} /Fe-N ₉₀₀ -PCC	150	8	7
3	Fe _{0.95} /Fe-P ₉₀₀ -PCC	150	5	5

Reaction conditions: 1 mmol quinoline, 100 mg catalyst, 2 mL solvent (heptane), 4 MPa H₂, 12 h. The conversion and yield were determined by GC using dodecane as an internal standard.

Table R2. Hydrogenation of nitrobenzene catalyzed by Fe_{1.0}/Fe-C₉₀₀-PCC, Fe_{1.0}/Fe-N₉₀₀-PCC and Fe_{0.95}/Fe-P₉₀₀-PCC.

Reaction scheme showing the hydrogenation of nitrobenzene to aniline using a catalyst and 4 MPa H₂.

Entry	Catalyst	Temperature (°C)	Conversion (%)	Yield (%)
1	Fe _{1.0} /Fe-C ₉₀₀ -PCC	100	9	8
2	Fe _{1.0} /Fe-N ₉₀₀ -PCC	100	12	12
3	Fe _{0.95} /Fe-P ₉₀₀ -PCC	100	11	10

Reaction conditions: 1 mmol nitrobenzene, 100 mg catalyst, 2 mL solvent (toluene), 4 MPa H₂, 18 h. The conversion and yield were determined by GC using n-hexadecane as an internal standard.

Reviewers' Comments:

Reviewer #1:

Remarks to the Author:

After the careful revising, I agree this paper should be published in Nature Commun. I do not have further serious comments, except that I would ask the authors to carefully check the references cited in this manuscript.

As one example, I found that in the rebuttal file, page 9, Reference 4. Hu, X. et al. Science. 363, 1091-1094 (2019).

I checked it should be vol. 364, not 363.

Reviewer #2:

Remarks to the Author:

In the revised manuscript, the author answered the questions we raised and gave detailed responses. However, before the manuscript being published in this high-standard journal, the following questions still need to be resolved.

1. Authors claim the heteroatom doping effect is the reason for decrease of the surface area of Fe-N900-PCC when compared with Fe-C900-PCC. However, in the corresponding references, the compounding of carbon nanotubes is the reason for increase of the surface area of GdFeO₃ + CNTs composites when compared with GdFeO₃. More convincing explanation is needed.

2. As shown in the Supplementary Table 6, the content of graphitic P in Fe-P900-PCC remains unchanged before and after hydrogenations (0.41 vs 0.41 atomic %). But why does the content of total P, C-O-P, C-PO₃/C₂-PO₂ changes significantly before and after the stability test?

Reviewer #3:

Remarks to the Author:

In this revised manuscript, the authors have performed additional experiments and DFT calculations to address all the comments raised previously. A much deeper insight of catalyst structure and reaction mechanism are provided. The manuscript is well written. Given the remarkable catalytic performance of the non-precious Iron single atom catalyst, this reviewer would recommend it for publication in Nature Communications.

Responses to Reviewers' Comments

Reviewer #1

Comment. *After the careful revising, I agree this paper should be published in Nature Communications. I do not have further serious comments, except that I would ask the authors to carefully check the references cited in this manuscript.*

As one example, I found that in the rebuttal file, page 9, Reference 4. Hu, X. et al. Science. 363, 1091-1094 (2019).

I checked it should be vol. 364, not 363.

Response. Thanks for your very careful review and suggestion. We have scrutinized the references and all of mistakes have been corrected.

Reviewer #2

Comment 1. In the revised manuscript, the author answered the questions we raised and gave detailed responses. However, before the manuscript being published in this high-standard journal, the following questions still need to be resolved.

Authors claim the heteroatom doping effect is the reason for decrease of the surface area of Fe-N₉₀₀-PCC when compared with Fe-C₉₀₀-PCC. However, in the corresponding references, the compounding of carbon nanotubes is the reason for increase of the surface area of GdFeO₃+CNTs composites when compared with GdFeO₃. More convincing explanation is needed.

Response 1. Thank you very much for this comment. Compared with Fe-C₉₀₀-PCC, both of BET specific surface area (768 cm²g⁻¹ to 628 cm²g⁻¹) and total pore volume (1.34 cm³g⁻¹ to 1.11 cm³g⁻¹) of Fe-N₉₀₀-PCC are decrease a little bit. On the other hand, the pore size distribution curves (Supplementary Figure 2b) demonstrate that these two catalysts have the same mean pore size (12.7 nm). Based on these results, we deduce that the difference in the number of mesopore leads to the difference of specific surface area and total pore volume of Fe-C₉₀₀-PCC and Fe-N₉₀₀-PCC. For the fabrication of Fe-N₉₀₀-PCC, cyanamide (N≡CNH₂, 5 g) is used as N precursor. Hence, the gas is inevitably produced by the decomposition of cyanamide in the carbonization process, then the gas release will result in the partial deterioration of mesopore structure^{1,2}. On the contrary, this phenomenon does not exist in the carbonization process of Fe-C₉₀₀-PCC, wherein mesopore structure derived from nanocasting of SiO₂ sphere is preserved well. Therefore, the number of mesopore in Fe-N₉₀₀-PCC is less than in Fe-C₉₀₀-PCC, eventually, the surface area and pore volume of Fe-N₉₀₀-PCC are lower than Fe-C₉₀₀-PCC.

References

1. Wang, C. et al. *Nano Energy*. **41**, 674-680 (2017).
2. Wang, Z. et al. *Microporous Mesoporous Mater.* **275**, 200-206 (2019).

Supplementary Figure 2b. Pore size distribution curves for Fe-C₉₀₀-PCC and Fe-N₉₀₀-PCC.

Figure R1. Schematic illustration of the preparation process of Fe-C₉₀₀-PCC and Fe-N₉₀₀-PCC.

Comment 2. As shown in the Supplementary Table 6, the content of graphitic P in Fe-P₉₀₀-PCC remains unchanged before and after hydrogenations (0.41 vs 0.41 atomic %). But why does the content of total P, C-O-P, C-PO₃/C₂-PO₂ changes significantly before and after the stability test?

Response 2. We appreciate the reviewer very much for this insightful comment. This is because the stability of graphitic P, C-O-P and C-PO₃/C₂-PO₂ are different. Graphitic P is generated by substitution of a graphitic carbon atom in the framework of graphite with a P atom, which can only be formed at higher temperature (>800 °C), so graphitic P is a kind of stable P species. On the contrary, C-O-P and C-PO₃/C₂-PO₂ groups are hanging on the surface or edge of graphite through C-P or O-P bond, they are unstable. Furthermore, the catalyst of Fe-P₉₀₀-PCC-H (Fe-P₉₀₀-PCC is treated with H₂ at 800 °C for 2 hours) have demonstrated that graphitic P cannot be removed by H₂ at 800 °C, meanwhile, the C-O-P and C-PO₃/C₂-PO₂ are easily destroyed under the same conditions. The content of C-O-P and C-PO₃/C₂-PO₂ changes significantly before and after the recyclability test is because they have been etched or decomposed under the hydrogenation reaction conditions. But the content of graphitic P keep constant before and after the recyclability test, this is why Fe-P₉₀₀-PCC has excellent stability.

Reviewer #3

Comment. *In this revised manuscript, the authors have performed additional experiments and DFT calculations to address all the comments raised previously. A much deeper insight of catalyst structure and reaction mechanism are provided. The manuscript is well written. Given the remarkable catalytic performance of the non-precious Iron single atom catalyst, this reviewer would recommend it for publication in Nature Communications.*

Response. Thank you very much for the positive comment.